# The formation of avian montane diversity across barriers and along elevational gradients

José Martín Pujolar[1,2], Mozes P. K. Blom [3,4], Andrew Hart Reeve[1], Jonathan D. Kennedy [1], Petter Zahl Marki[1], Thorfinn S. Korneliussen[5], Benjamin G. Freeman [6], Katerina Sam[7,8], Ethan Linck[9], Tri Haryoko [10], Bulisa Iova[11], Bonny Koane[12], Gibson Maiah[12], Luda Paul[12], Martin Irestedt [3] & Knud Andreas Jønsson [1✉]

Tropical mountains harbor exceptional concentrations of Earth's biodiversity. In topographically complex landscapes, montane species typically inhabit multiple mountainous regions, but are absent in intervening lowland environments. Here we report a comparative analysis of genome-wide DNA polymorphism data for population pairs from eighteen Indo-Pacific bird species from the Moluccan islands of Buru and Seram and from across the island of New Guinea. We test how barrier strength and relative elevational distribution predict population differentiation, rates of historical gene flow, and changes in effective population sizes through time. We find population differentiation to be consistently and positively correlated with barrier strength and a species' altitudinal floor. Additionally, we find that Pleistocene climate oscillations have had a dramatic influence on the demographics of all species but were most pronounced in regions of smaller geographic area. Surprisingly, even the most divergent taxon pairs at the highest elevations experience gene flow across barriers, implying that dispersal between montane regions is important for the formation of montane assemblages.

[1] Natural History Museum of Denmark, University of Copenhagen, Universitetsparken 15, DK-2100 Copenhagen, Denmark. [2] Centre for Ocean Life, DTU Aqua, Kemitorvet, building 202, DK-2800 Kgs Lyngby, Denmark. [3] Department of Bioinformatics and Genetics, Swedish Museum of Natural History, SE-104 05 Stockholm, Sweden. [4] Museum für Naturkunde, Leibniz Institut für Evolutions- und Biodiversitätsforschung, Berlin, Germany. [5] Lundbeck Foundation GeoGenetics Center, Globe Institute, University of Copenhagen, Copenhagen, Denmark. [6] Biodiversity Research Centre, University of British Columbia, Vancouver, BC, Canada. [7] Biology Centre of Czech Academy of Sciences, Institute of Entomology, Branisovska 31, Ceske Budejovice, Czech Republic. [8] University of South Bohemia, Faculty of Science, Branisovska 1760, Ceske Budejovice, Czech Republic. [9] Department of Biology & Museum of Southwestern Biology, University of New Mexico, Albuquerque, NM, USA. [10] Museum Zoologicum Bogoriense, Research Center for Biology, the National Research and Innovation Agency (BRIN), Jl. Raya Jakarta-Bogor Km 46, Cibinong 16911, Indonesia. [11] Papua New Guinea National Museum and Art Gallery, Port Moresby, Papua New Guinea. [12] The New Guinea Binatang Research Centre, Madang, Papua New Guinea. ✉email: kajonsson@snm.ku.dk

A fundamental goal of biology is to understand the mechanisms that underlie the spatio-temporal formation of species and communities. Divergence among geographically isolated populations is the common mechanism by which speciation is initiated in vertebrate taxa[1–4]. Physical barriers that differ in their capacity to maintain the geographic separation of populations are likely to directly impact gene flow, population differentiation and ultimately, speciation[1,2,5]. Alternatively, local adaptation to particular environments across gradients can generate diversity in the absence of physical barriers[6–12]. Because the efficacy of a given barrier or gradient is mediated by the distribution of the species that encounters it, evaluating its overall influence on evolutionary processes requires comparative data from geographically isolated taxon pairs that occur across gradients in different environmental settings, and that possess a variety of ecological characters. Such data can provide estimates of when populations become geographically isolated from one another, the extent of gene flow between incipient species (diverging populations) and temporal changes in population demography. Here we integrate these estimates to ask: (i) Does barrier strength influence population differentiation? (ii) Does barrier strength affect population differentiation of species with different elevational distributions? (iii) To what extent do population sizes fluctuate over time, (iv) Is there evidence of population segregation along elevational gradients? (v) Are montane populations recruited from the lowlands or do they arise through colonization from mountaintop to mountaintop? (vi) How do levels and directionality of gene flow (i.e. dispersal) vary between populations of species across elevational gradients?

Answering these questions allow us to address three competing explanations of the evolutionary origin of montane populations. First, populations that currently live only at high elevations may have originated in the lowlands but shifted upslope over time[13–15]. This would lead to a signature of populations of species that inhabit higher elevations being more differentiated compared to populations of species inhabiting lower elevation. Second, populations may colonize one montane region from another montane region[15]. For example, in New Guinea, montane taxa might repeatedly colonize uni-directionally from the dominant central mountain range to the many smaller outlying ranges. If this is the case, we expect to find evidence of gene flow even between montane populations. Third, high elevation populations may form by in situ parapatric speciation, i.e. through the break-up of one population into two along an elevational gradient for example through adaptation to specific elevational environments. This would be evidenced by high levels of differentiation between individuals of the same population that occur at high elevations and lower elevations.

For this study, we use explicit demographic models to analyse genomic data of discrete populations with different elevational distributions. We assembled 15 de novo genomes and generated whole-genome resequencing data for eighteen forest bird species with populations distributed along multiple elevational gradients, geographically separated by distinct barriers (Fig. 1). We focus on two pairs of elevational gradients in distinct geographic settings: (i) the neighbouring montane islands of Buru and Seram in the Moluccas of Indonesia, and (ii) two mountain ranges in New Guinea separated by a lowland barrier. The two oceanic Moluccan islands are separated by a "hard" deep-sea barrier; and are relatively symmetric in terms of age (3 Mya)[16] and area (≈ 4000 km² above 500 m and ≈1000 km² above 1000 m). The two focal mountain ranges in New Guinea, on the other hand, are part of continental Australo-Papua. They are separated by a "soft" lowland barrier; and are asymmetric in terms of age and area. While estimates of the age of the New Guinean terranes vary significantly, the large Central Range (≈150,000 km² above

1000 m) started to rise some 10 Mya and reached its present elevation about 5 Mya, while the Huon mountains (≈10,000 km² above 1000 m) rose above sea-level some 4 Mya and reached their present elevation about 1.5 Mya[17,18]. The mountains of New Guinea reach > 4000 m a.s.l, while the mountains on Buru and Seram are lower, reaching 2500 m a.s.l. and 3000 m a.s.l., respectively. However, due to the Massenerhebung effect[19], temperature drops more quickly with increasing altitude in the isolated Moluccan islands than in New Guinea, and both systems encompass environmental extremes from sweltering lowland swamps to chilly mountain cloud forests. Finally, while the geological history has affected the Indo-Pacific biota significantly, dramatic climatic oscillations during the Pleistocene are also expected to impact diversity build-up in the region[20,21].

Our analyses of population divergence across two montane systems show that: (i) Genetic differentiation is generally greater across the ocean barrier than the lowland barrier. (ii) Differentiation is greater between populations of montane species than between populations of lowland species in New Guinea but not in the Moluccas. (iii) There were marked fluctuations in population sizes during the Pleistocene. (iv) There is negligible differentiation along elevational gradients, hence little evidence for sympatric speciation. (v) Individuals of montane populations continuously disperse to other mountains. (vi) The direction of colonization is predominantly from a larger mountain range to a smaller mountain range.

## Results

**Genome sequencing and assembly**. Genome assemblies ranged in size from 799.9 Mbp in *Melanocharis versteri* to 1053.5 Mbp in *Sericornis nouhuysi*. The number of scaffolds ranged from 14,086 scaffolds in *Melipotes ater* to 87,957 scaffolds in *Ficedula hyperythra* and N50 ranged between ca. 40 Kbp to and 25 Mbp. Benchmarking Universal Single-Copy Orthologs (BUSCO) analyses of genome completeness ranged from a high proportion of complete BUSCOs in *Melipotes ater*, 86.8% to only 66.7% complete BUSCOs in *Rhipidura albolimbata*. For most species, the proportions of complete BUSCOs were 75–80%. Overall, the proportion of missing BUSCOs was low, ranging from 6.6% in *Melipotes ater* to 15.2% in *Rhipidura albolimbata* (see Supplementary Table 1 for all genome assembly statistics and Supplementary Fig. 1 for the number of SNP variants per species).

**Kinship analyses of individuals within populations**. Sampling of closely related individuals can dramatically bias estimates of population structure and demographics. Two *Pachycephala schlegelii* individuals (A117 and A118) showed a pairwise kinship coefficient of 0.144, indicative of being half-siblings. The two individuals were collected at the same locality on the same date. Similarly, two *Ifrita kowaldi* individuals (D116 and D117) showed a pairwise kinship coefficient of 0.135, also suggestive of being half-siblings. In this case, the individuals were collected on the same sampling locality on two consecutive days. To not bias downstream demographic analyses, one of the *P. schlegelii* (A118) and one of the *I. kowaldi* (D117) individuals were excluded from all subsequent analyses. For all other species, no closely related individuals were identified.

**Genetic differentiation**. Estimated levels of differentiation between populations were initially based on three approaches; (i) calculation of $F_{ST}$ (the fixation index), which quantifies the degree of genetic differentiation between populations based on the variation in allele frequencies, ranging between 0 (complete mixing of individuals) and 1 (complete separation of populations) (Fig. 1), (ii) Standardized pairwise $F_{ST}$ used to conduct a Principal

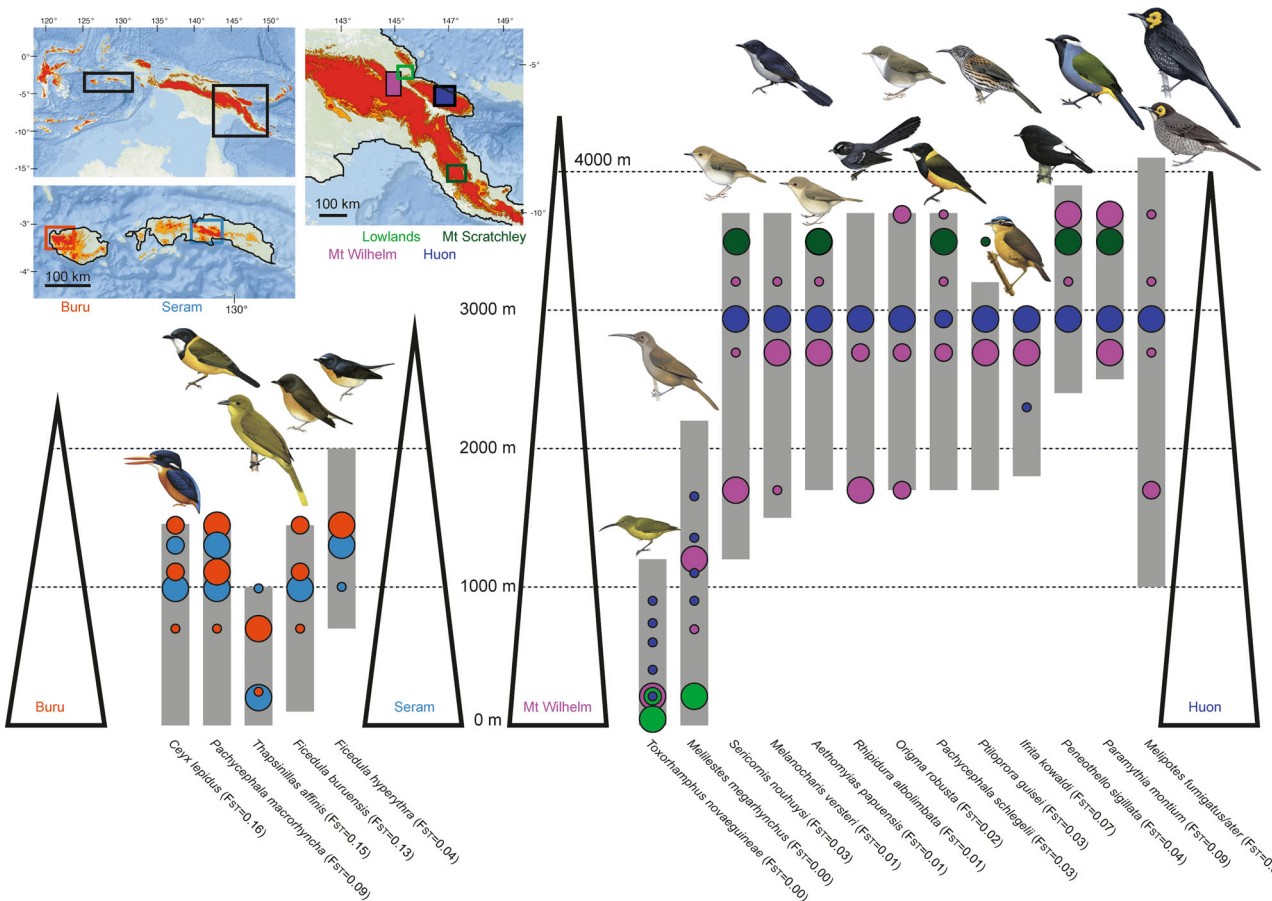

**Fig. 1 Overview of sampling localities, elevations and species included in the study.** Buru and Seram samples to the left and New Guinean samples to the right. The maximum elevation of Mount Scratchley (not illustrated) is 4072 m a.s.l. Grey columns illustrate elevational distributions from the literature and the size of the circles indicate the number of individuals sampled at particular elevations (small = 1 individual, medium = 2 individuals, large ≥ 3 individuals). Circles from Buru are red and circles from Seram are light blue. For New Guinea, circles from Mount Wilhelm are purple, circles from Huon are dark blue, circles from Mount Scratchley are dark green and circles from the lowlands just north of the Ramu/Markham River are light green. Illustrations of the focal bird species from del Hoyo et al.[68].

Component Analysis (PCA) in order to visualize population structure (Supplementary Fig. 1) and (iii) Admixture analysis as implemented in STRUCTURE (a clustering algorithm that infers the most likely number of groups [K]), in which individuals are grouped into clusters according to the proportion of their ancestry components (Supplementary Fig. 1). As a preliminary analysis, we calculated $F_{ST}$ and constructed PCA plots for the four congeneric (incl. *Sericornis/Aethomyias* [until recently placed in the genus *Sericornis*]) species pairs in our study (Supplementary Fig. 2), which were aligned using the same reference genome. This was done to ascertain that no samples had been misidentified and to gauge levels of differentiation between distinct species. All species were genetically well separated and $F_{ST}$ values ranged from 0.08 for the two *Ptiloprora* species to 0.20 for the two *Ficedula* species.

For five out of six species from Buru/Seram, genetic differentiation ($F_{ST}$) was high between islands (Fig. 1), and comparable to differentiation between named congeneric species in this study (e.g. *Ptiloprora* and *Melipotes*); *Ceyx lepidus* ($F_{ST} = 0.16$), *Thapsinillas affinis* ($F_{ST} = 0.15$), *Ficedula buruensis* ($F_{ST} = 0.13$) and *Pachycephala macrorhyncha* ($F_{ST} = 0.09$). In contrast, differentiation in *Ficedula hyperythra* was consistent with population-level differentiation ($F_{ST} = 0.04$). In all cases, individuals from Buru and Seram were clearly differentiated in the PCA and STRUCTURE plots (Supplementary Fig. 1A). For *Ceyx lepidus*, *Ficedula buruensis* and *Pachycephala macrorhyncha*,

samples were collected at multiple elevations and we therefore calculated genetic differentiation between elevations (Buru: 1097 m versus 1435 m and Seram: 1000 m versus 1300 m) to determine any potential parapatric differentiation along the gradients. In all possible comparisons, $F_{ST}$ values did not differ significantly from 0. Moreover, PCA plots showed that samples did not cluster according to elevation (Supplementary Fig. 3A).

Three of the thirteen New Guinean population pairs occurring in Mount Wilhelm and Huon showed relatively high genetic divergences: *Melipotes fumigatus/ater* ($F_{ST} = 0.08$), *Paramythia montium* ($F_{ST} = 0.09$) and *Ifrita kowaldi* ($F_{ST} = 0.07$) (Fig. 1) with populations clearly separated (Supplementary Fig. 1). By contrast, the two lowland species *Toxorhamphus novaeguineae* and *Melilestes megarhynchus* showed little genetic differentiation, $F_{ST} = 0.00$. For the remaining species, genetic differentiation between Mount Wilhelm and Huon ranged between $F_{ST} = 0.01$–$0.05$. Despite this moderate level of genetic differentiation, the populations of Mount Wilhelm and Huon could be clearly distinguished in the PCA plots. In all cases STRUCTURE suggested a scenario with K = 2 with some mixing of individuals, except for *Rhipidura albolimbata*, in which K = 1 was suggested.

For five bird species we included an additional population from Mount Scratchley, which is also situated in the Central Range but ~400 km to the southeast of Mount Wilhelm. Genetic differentiation of this population from the other two populations was comparable with that between Mount Wilhelm and Huon. The

highest genetic differentiation was found in *Paramythia montium* ($F_{ST} = 0.10$ both between Mount Wilhelm and Mount Scratchley and between Huon and Mount Scratchley). In the case of *Peneothello sigillata*, the Mount Scratchley population appeared genetically well-differentiated from both the populations of Mount Wilhelm ($F_{ST} = 0.06$) and Huon ($F_{ST} = 0.07$). In both cases, STRUCTURE suggested a scenario of K = 3, with individual assignments matching the three geographically circumscribed populations. For *Pachycephala schlegelii*, genetic differentiation was relatively high between Huon and Mount Scratchley ($F_{ST} = 0.05$), but low between Mount Wilhelm and Mount Scratchley ($F_{ST} = 0.01$). Accordingly, STRUCTURE suggested a scenario with K = 2 groups. For the remaining two species *Sericornis nouhuysi* showed some differentiation ($F_{ST} = 0.03$) between Mount Wilhelm and Huon and *Aethomyias papuensis* showed minor differentiation ($F_{ST} = 0.02$ between Mount Scratchley and Huon (Supplementary Table 2), but for both species, STRUCTURE suggested a scenario of K = 2 with considerable mixing of individuals between populations.

Samples from Mount Wilhelm were collected at elevations ranging from 1700 to 3700 m, again allowing us to test for differences within populations on a single slope, a finding that would be consistent with incipient parapatric speciation. No species showed significant differences in $F_{ST}$ when comparing individuals from different elevations, and concordantly there was little clustering of individuals by elevation in the PCA plots. Even when individuals were collected as far as 2000 elevational meters apart (as in the case of *Origma robusta*), genetic differentiation was low ($F_{ST} = 0.01$). In Huon, all samples were collected at the same elevation, except for *Ifrita kowaldi*, for which genetic differentiation of $F_{ST} = 0.03$ was found between individuals collected at 2300 m and 2950 m (Supplementary Fig. 3B, Supplementary Table 2). These analyses however, suffer from very small sample sizes that hinder a thorough analysis of parapatric speciation events. Furthermore, we note that divergence with gene flow may not manifest as a genome-wide phenomenon (at least, not until the taxa are so differentiated that gene flow has ceased). Instead, it may proceed via selection acting to create small 'islands of differentiation' within the genome against a background of negligible differentiation[22,23]. Such analyses require large sample sizes and are therefore not possible herein.

**Correlations between genetic divergence and elevation.** If lineages colonize mountains from the lowlands, followed by range contraction and differentiation in the highlands, we would expect a signature of larger genetic differentiation ($F_{ST}$) between populations inhabiting higher elevations. We found no relationship between genetic differentiation ($F_{ST}$) and the altitudinal floor (the lowest elevation at which a species/population occurs) for the five Moluccan species, but for all New Guinean taxa with the exception of *Melipotes fumigatus/ater* we found a significant positive correlation ($r = 0.83$, $p < 0.001$). This relationship remained significant even when excluding the two lowland taxa (*Toxorhamphus novaeguineae* and *Melilestes megarhynchus*, $r = 0.70$, $p = 0.022$).

**Demographic inferences using Pairwise Sequentially Markovian Coalescent (PSMC) analyses.** The demographic history of the eighteen bird species in our study was first inferred by analyzing the whole-genome sequence of one single individual per species using Pairwise Sequentially Markovian Coalescent (PSMC)[24] (Supplementary Fig. 4). PSMC relies on the distribution of heterozygous sites across the genome and infers the distribution of the time since the most recent common ancestor

(TMRCA) between each pair of alleles at all loci across the whole genome of one single individual. This provides information about how effective population sizes change over time.

The inferred demographic histories of the eighteen bird species encompassed the time period from ca. 10 Mya to 20 Kya (note that most of the species included herein are younger than 10 My and as such past demographic dynamics reflect an ancestral form, likely with a very different distribution, Supplementary Fig. 4). Initial population sizes for most species were ca. 300,000–400,000 individuals, with the lowest initial population size being *Ifrita kowaldi* with ca. 200,000 individuals, and the highest being *Toxorhamphus novaeguineae* and *Peneothello sigillata* with ca. 1,000,000 individuals.

Inferred demographic fluctuations over time show that most species had an initial period of relative demographic stability during the Miocene (5.3–23 Mya) and Pliocene (2.6–5.3 Mya). During this period, most species showed a rather flat curve. The most dramatic demographic changes for the majority of species occurred in the Pleistocene (beginning around 2.6 Mya) but the timing of population increases and decreases vary between species. All PSMC results were consistent when bootstrapping the data but clearly become more dubious toward the present (Supplementary Fig. 4).

PSMC plots generated for the re-sequenced individuals (Figs. 2–4) allowed us to compare the demographic histories of the different populations—those on Mount Wilhelm, Huon, and Mount Scratchley in New Guinea; and on Buru and Seram in the Moluccas. These plots closely mimicked the demographic changes inferred from the de novo assembled reference sequences (Supplementary Fig. 4). Individuals from the same population showed largely similar demographic trajectories and only differed for the most recent time periods. For three New Guinean species (*Melanocharis versteri*, *Toxorhamphus novaeguineae* and *Melilestes megarhynchus*), populations from Mount Wilhelm and Huon shared the same demographic history over time. However, for all other New Guinean species, populations from Mount Wilhelm and Huon showed contrasting demographic patterns from the mid-late Pleistocene towards the present: populations from Huon experienced an important demographic decline, dropping to a historical low by the Late Pleistocene, while populations from Wilhelm remained stable and did not experience the decline observed in Huon. This contrasting demographic pattern can be mostly clearly observed in *Origma robusta*, *Ifrita kowaldi*, *Rhipidura albolimbata*, *Ptiloprora guisei* and *Melipotes fumigatus/ater*. Similarly, populations of the species occurring on Mount Wilhelm, Huon and Mount Scratchley showed comparable demographic histories up until the mid-late Pleistocene, at which point each population began to behave slightly differently, with Huon always showing the lowest population sizes.

Of the species occurring in the Moluccas, *Ficedula hyperythra* and *Thapsinillas affinis* showed a common demographic history on both Buru and Seram. The two *Ceyx lepidus* populations also appear to have a similar demographic history up until 1 Mya, when the effective population size spiked on Buru, but not on Seram (Fig. 2).

For the three species with no de novo genome (*Aethomyias papuensis*, *Pachycephala macrorhyncha* and *Ficedula buruensis*), we conducted PSMC analyses with re-sequenced data mapped to the following non-conspecific reference genomes: *Sericornis nouhuysi* (for the closely related *Aethomyias papuensis*), *Pachycephala schlegelii* (for the closely related *Pachycephala macrorhyncha*) and *Ficedula hyperythra* (for the closely related *Ficedula buruensis*). While the PSMC plots for these three species should be interpreted with some caution, we observed the same large population expansions during the Pleistocene, but the expansion occurred in the late- rather than the mid-Pleistocene.

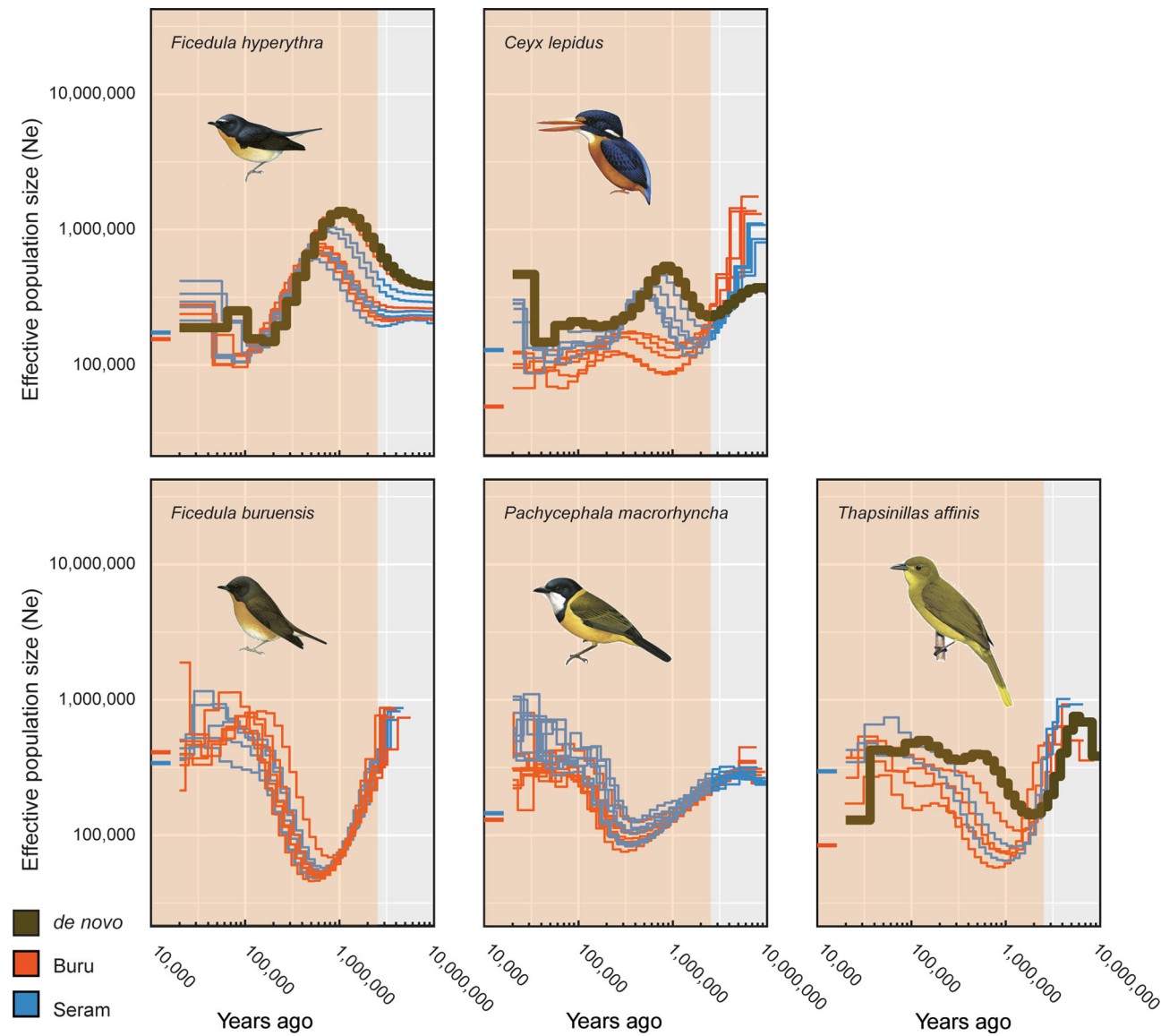

**Fig. 2 PSMC estimates of the demographic changes (Ne = effective population size) over time for individuals of five bird species with two populations sampled from the Moluccas.** The brown curve is the PSMC estimate for the de novo sequence data (for PSMC estimates for 100 bootstrapped sequences see Supplementary Fig. 4). The PSMC estimates for the resequencing data are presented for *Ficedula buruensis, Ficedula hyperythra, Pachycephala macrorhyncha, Thapsinillas affinis* and *Ceyx lepidus*) with populations from Buru (red) and Seram (blue). The most recent 20 Ky have been removed from the PSMC plots and the current Ne estimate indicated at T = 10 Kya is based on the demographic analyses in fastsimcoal2. The shaded area indicates the Pleistocene. Illustrations of the focal bird species from del Hoyo et al.[68].

Overall, the PSMC plots obtained from resequencing data (Figs. 2–4) were largely concordant with the PSMC plots obtained from de novo sequencing (Supplementary Fig. 4). However, plots from resequencing data differed in showing variable trajectories from 20 to 30,000 years ago to the present, suggesting that recent demographic changes and current population sizes inferred with PSMC are uncertain and should be interpreted with caution. In contrast, the demographic histories of the species between further back in time appeared robust, with good agreement between the PSMC plots derived from the de novo and resequencing data.

**Demographic inference using explicit model testing**. The demographic history of the populations of pairs of the eighteen bird species in our study was further investigated using the model-testing approach implemented in fastsimcoal2[25]. We present a summary of the results in Table 1 and the results from all models tested in Supplementary Data 1, including current and

ancestral population sizes, migration rates and divergence times and summarize the major findings below.

Models including migration always performed better than models without migration. Migration rates between Buru and Seram were low for the five species compared, ranging between proportions of $10^{-6}$ and $10^{-7}$ per generation. Migration rates between Mount Wilhelm and Huon were more variable, and while low migration rates (ca. $10^{-6}$–$10^{-7}$) were suggested for those species with high genetic differentiation (i.e. *Melipotes fumigatus/ater* and *Ifrita kowaldi*), higher migration rates (ca. $10^{-5}$) were suggested for the lowland species *Toxorhamphus novaeguineae* and *Melilestes megarhynchus* and for those species with low genetic differentiation (e.g. *Melanocharis versteri, Rhipidura albolimbata* and *Ptiloprora guisei*). For a few species, migration rates were close to zero ($10^{-11}$–$10^{-18}$), but we found no clear directionality in migration rates between Mount Wilhelm and Huon.

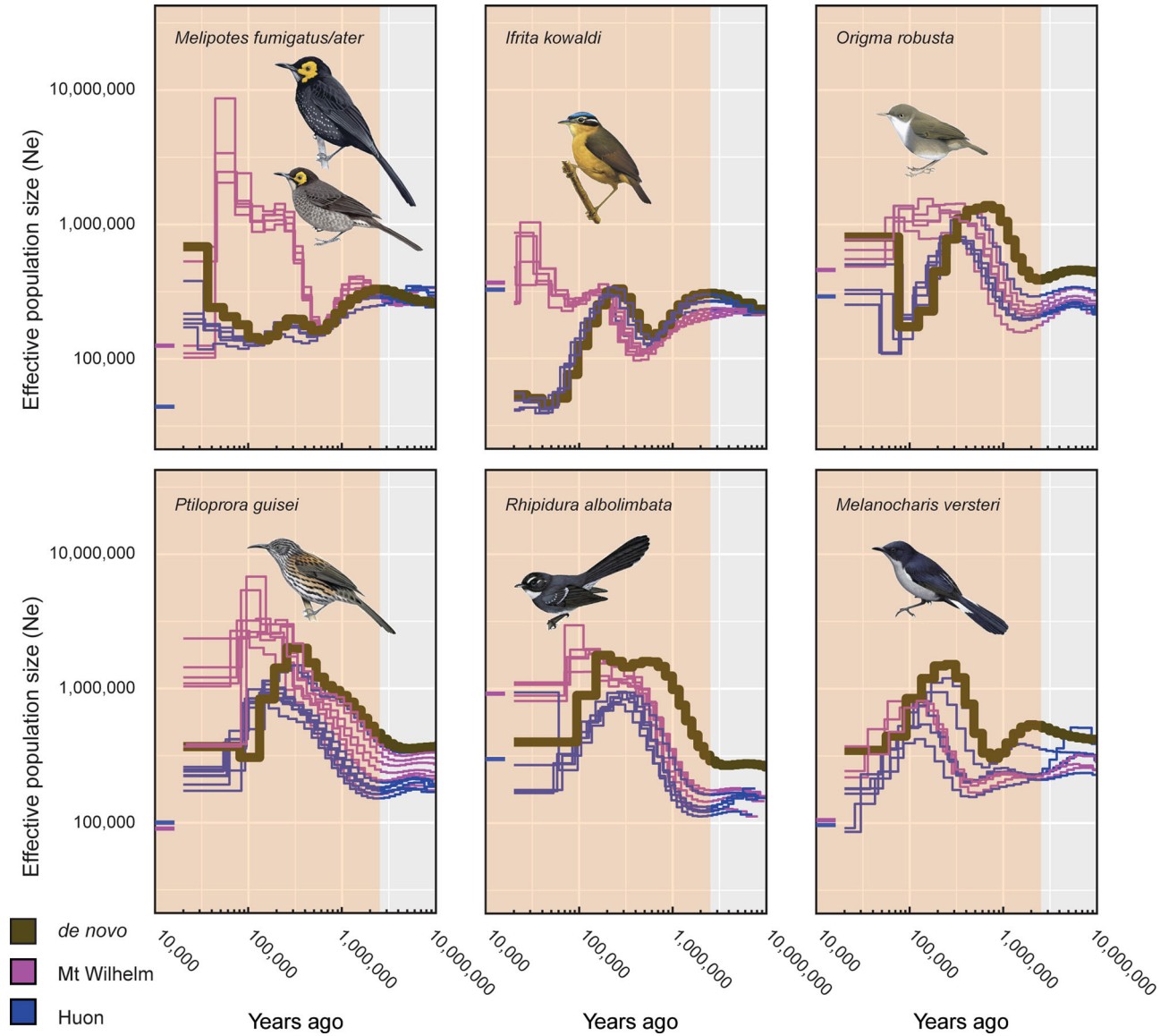

**Fig. 3 PSMC estimates of the demographic changes (Ne = effective population size) over time for individuals of six bird species with two populations sampled from New Guinea.** The brown curve is the PSMC estimate for the de novo sequence data (for PSMC estimates for 100 bootstrapped sequences see Supplementary Fig. 4). The PSMC estimates for the resequencing data are presented for *Melipotes fumigatus/ater*, *Ifrita kowaldi*, *Origma robusta*, *Ptiloprora guisei*, *Rhipidura albolimbata* and *Melanocharis versteri* with populations from Mount Wilhelm (purple) and Huon (blue). The most recent 20 Ky have been removed from the PSMC plots and the current Ne estimate indicated at T = 10 Kya is based on the demographic analyses in fastsimcoal2. The shaded area indicates the Pleistocene. Illustrations of the focal bird species from del Hoyo et al.[68]

Ancestral population sizes were similar across species, with an average of ca. 325,000 individuals, ranging from ca. 100,000 in *Thapsinillas affinis* to ca. 600,000 individuals in *Ficedula buruensis* (see Supplementary Data 1 for confidence intervals). Current population sizes for the New Guinean species were generally higher in Mount Wilhelm than Huon, with an average population size of ca. 400,000 individuals per species for the former, and an average of ca. 200,000 individuals for the latter. Current population sizes ranged from ca. 75,000 individuals in *Aethomyias papuensis* to ca. 900,000 individuals in *Rhipidura albolimbata* in Mount Wilhelm, and from ca. 35,000 individuals in *Paramythia montium* to ca. 700,000 in *M. megarhynchus* in Huon. Current population sizes in Mount Scratchley were also generally lower than in Mount Wilhelm, but similar to the values found in Huon, ranging from ca. 25,000 individuals in *Peneothello sigillata* to ca. 400,000 individuals in *Pachycephala schlegelii*. For the Moluccan species, current population sizes were higher in Seram, with an

average of ca. 225,000 individuals per species, relative to Buru, with an average of ca. 175,000 individuals. Current population sizes ranged from ca. 140,000 individuals in *Ceyx lepidus* to ca. 350,000 in *Ficedula buruensis* in Seram, and from 50,000 individuals in *Ceyx lepidus* to 500,000 in *Ficedula buruensis* in Buru. Divergence times between populations in the Moluccan islands ranged between 0.1 and 1.4 My (mean = 0.583 My) and between 0.1 and 1.8 My (mean = 1.2 My) for the New Guinean populations.

## Discussion

**Does barrier strength predictably influence differentiation of disjunct populations?** Across the two montane systems compared in our study, genetic differentiation is generally higher between populations of the species distributed on separate Moluccan islands (separated by a "hard" oceanic barrier) than in those found on different New Guinean mountain ranges

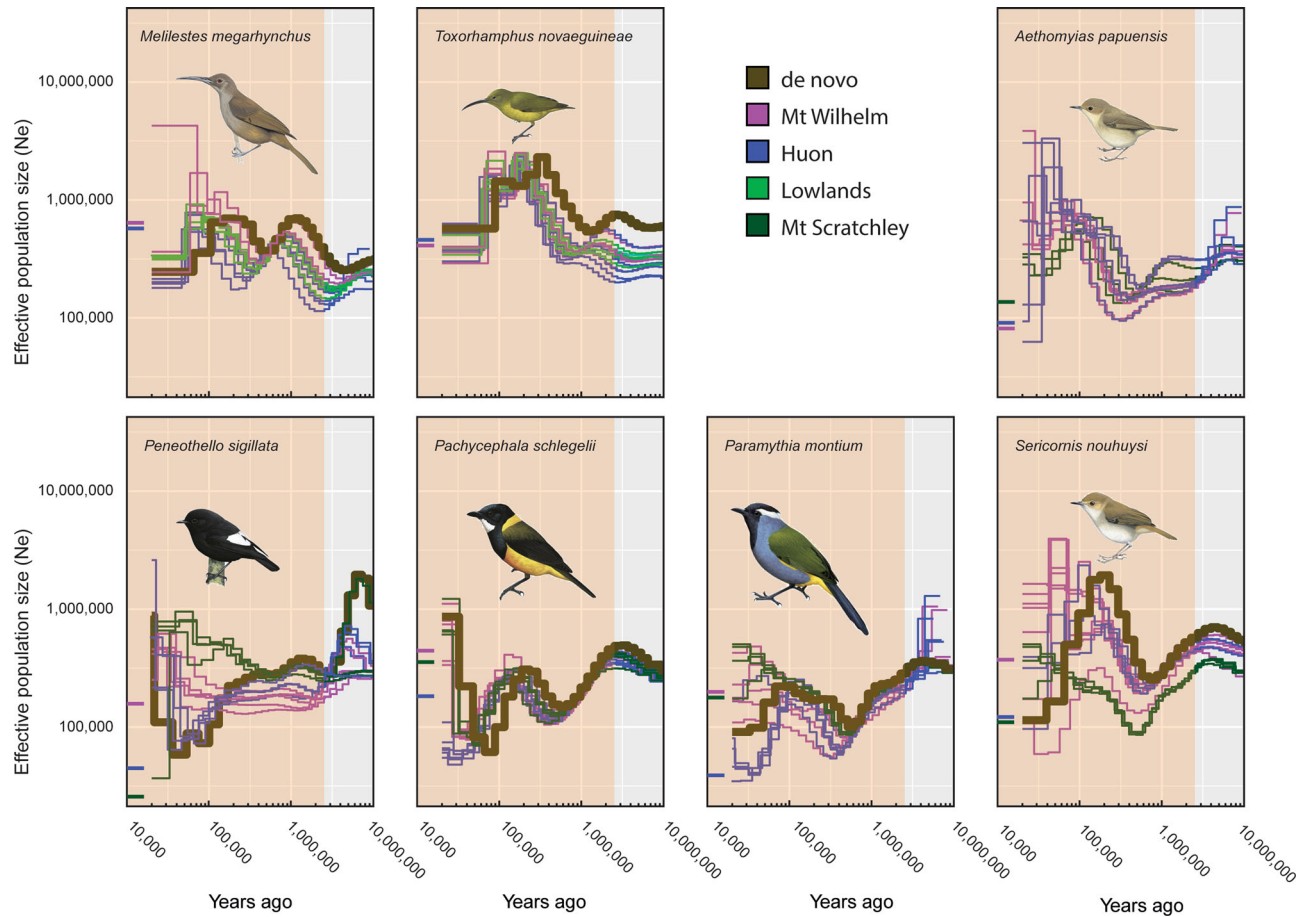

**Fig. 4 PSMC estimates of the demographic changes (Ne = effective population size) over time for individuals of seven bird species with three populations sampled from New Guinea.** The brown curve is the PSMC estimate for the de novo sequence data (for PSMC estimates for 100 bootstrapped sequences see Supplementary Fig. 4). The PSMC estimates for the resequencing data are presented for *Melilestes megarhynchus*, *Toxorhamphus novaeguineae* including a third population from the lowlands north of the Ramu/Markham River (light green) and New Guinean species (*Aethomyias papuensis*, *Peneothello sigillata*, *Pachycephala schlegelii*, *Paramythia montium* and *Sericornis nouhuysi*) including a third montane population from Mount Scratchley (dark green) in the south-eastern central range. Populations from Mount Wilhelm (purple) and Huon (blue). The most recent 20 Ky have been removed from the PSMC plots and the current Ne estimate indicated at T = 10 Kya is based on the demographic analyses in fastsimcoal2. The shaded area indicates the Pleistocene. Illustrations of the focal bird species from del Hoyo et al.[68].

(separated by a "soft" lowland barrier). Four of the five species examined in the Moluccas have genetically well-separated populations ($F_{ST}$ = 0.09–0.16) on the two islands of Buru and Seram (Fig. 1 and Supplementary Fig. 1). This is corroborated by recent analyses based on mitochondrial[26,27], acoustic and morphological data[28,29]. Only *Ficedula hyperythra*, which is found at higher elevations on both islands than its congener *F. buruensis*, shows comparatively low levels of differentiation between Buru and Seram ($F_{ST}$ = 0.04). *Ficedula hyperythra* appears to have undergone a broad and recent radiation from mainland Asia across the mountains of the Indonesian islands[30]. That *F. hyperythra* has a higher dispersal capacity (>1 dispersing individuals per generation between populations) than the other focal Moluccan species (<1 dispersing individuals per generation between populations) is supported by dispersal estimates using fastsimcoal2 (Table 1), and this explains the lower genetic differentiation found between populations. Overall, our data reinforce the notion that the avifaunas of Buru and Seram have always been well isolated from one another by a deep-sea barrier. Following colonization of the islands of an ancestral taxon, populations on the two islands would quickly embark on independent evolutionary trajectories. And today the two islands support unique avifaunas with many endemic species[29,31–34].

Populations of the thirteen species sampled from New Guinea's large Central Range (Mount Wilhelm and Mount Scratchley) and the smaller Saruwaged Range on the Huon Peninsula represent members of the island's core passerine bird clades (Supplementary Fig. 5) and range across various elevational zones from sea-level to the treeline (3700 m a.s.l.). The two montane species of *Melipotes* honeyeaters, one endemic to Huon and the other widespread across the Central Range, were included as a reference point for other analyses as they are closely related, but clearly represent distinct species[35,36] and are genetically well-differentiated ($F_{ST}$ = 0.08, Supplementary Fig. 1A). In both *Paramythia montium* and *Ifrita kowaldi*, genetic differentiation between Huon and Mount Wilhelm populations was as high as that found between the two *Melipotes* species (Fig. 1). This is unexpected given the respective populations' mitochondrial homogeneity[37] and virtually identical plumage and ecology[38]. Populations of the eight remaining New Guinean highland species showed weak differentiation between the Central Range and Huon ($F_{ST}$ = 0.01–0.04, Fig. 1). In contrast, populations of the two lowland species (*Melilestes megarhynchus* and *T. novaeguineae*, Fig. 1) are completely undifferentiated ($F_{ST}$ = 0.00), and mix freely across the intervening lowlands. The high gene flow predicted between populations of the lowland species is concordant with the dispersal estimates (3–14 dispersing individuals per generation between populations) in fastsimcoal2 (Table 1 and Supplementary Data 1). Dispersal estimates

**Table 1 Summary of results from fastsimcoal2 including estimated values for current effective population sizes (Ne) in Buru, Seram, Mount Wilhelm, Huon and Mount Scratchley, ancestral effective population size (Ne ancestral), divergence time (T DIV), effective migration rates per generation (M) between populations, and effective number of migrants per generation (Nm) between populations (for more details see Supplementary Data 1).**

| Two populations (Buru/ Seram) | Ne Buru | Ne Seram | Ne ancestral | T DIV (y) | M Buru to Seram | M Seram to Buru | Nm Buru to Seram | Nm Seram to Buru |
|---|---|---|---|---|---|---|---|---|
| Ceyx lepidus | 50,000 | 140,000 | 510,000 | 1,400,000 | $2 \times 10^{-7}$ | $3 \times 10^{-6}$ | 0.0 | 0.4 |
| Ficedula buruensis | 480,000 | 350,000 | 610,000 | 350,000 | $4 \times 10^{-7}$ | $2 \times 10^{-7}$ | 0.2 | 0.1 |
| Ficedula hyperythra | 150,000 | 180,000 | 290,000 | 1,400,000 | $9 \times 10^{-6}$ | $6 \times 10^{-6}$ | 1.4 | 1.1 |
| Pachycephala macrorhyncha | 140,000 | 160,000 | 310,000 | 100,000 | $2 \times 10^{-6}$ | $1 \times 10^{-6}$ | 0.3 | 0.2 |
| Thapsinillas affinis | 80,000 | 270,000 | 110,000 | 250,000 | $2 \times 10^{-6}$ | $1 \times 10^{-6}$ | 0.2 | 0.3 |

| 2 populations (Wilhelm/ Huon) | Ne Wilhelm | Ne Huon | Ne ancestral | T DIV (y) | M Wilhelm to Huon | M Huon to Wilhelm | Nm Wilhelm to Huon | Nm Huon to Wilhelm |
|---|---|---|---|---|---|---|---|---|
| Melipotes fumigatus/ater | 140,000 | 50,000 | 330,000 | 100,000 | $4 \times 10^{-6}$ | $2 \times 10^{-6}$ | 0.6 | 0.1 |
| Ifrita kowaldi | 420,000 | 360,000 | 400,000 | 500,000 | $9 \times 10^{-7}$ | $5 \times 10^{-6}$ | 0.4 | 1.8 |
| Melanocharis versteri | 110,000 | 90,000 | 420,000 | 1,800,000 | $9 \times 10^{-5}$ | $2 \times 10^{-5}$ | 10.0 | 1.8 |
| Origma robusta | 500,000 | 290,000 | 200,000 | 1,000,000 | $4 \times 10^{-6}$ | $6 \times 10^{-6}$ | 2.0 | 1.7 |
| Ptiloprora guisei | 90,000 | 100,000 | 300,000 | 1,400,000 | $3 \times 10^{-5}$ | $1 \times 10^{-5}$ | 2.7 | 1.0 |
| Rhipidura albolimbata | 910,000 | 320,000 | 170,000 | 1,050,000 | $1 \times 10^{-5}$ | $6 \times 10^{-6}$ | 9.1 | 2.0 |
| Meliestes megarhynchus | 740,000 | 710,000 | 360,000 | 1,400,000 | $4 \times 10^{-6}$ | $2 \times 10^{-5}$ | 3.0 | 14.2 |
| Toxorhamphus novaeguineae | 470,000 | 520,000 | 380,000 | 1,900,000 | $1 \times 10^{-5}$ | $1 \times 10^{-5}$ | 4.7 | 5.2 |

| 3 populations (Wilhelm/ Huon/ Scratchley) | Ne Wilhelm | Ne Huon | Ne Scratchley | Ne ancestral | T DIV (y) | M Wilhelm to Huon | M Huon to Wilhelm | M Wilhelm to Scratchley | M Scratchley to Wilhelm | M Huon to Scratchley | M Scratchley to Huon |
|---|---|---|---|---|---|---|---|---|---|---|---|
| Pachycephala schlegelii | 510,000 | 210,000 | 400,000 | 410,000 | 900,000 | $1 \times 10^{-5}$ | $5 \times 10^{-11}$ | $8 \times 10^{-13}$ | $2 \times 10^{-10}$ | $3 \times 10^{-6}$ | $3 \times 10^{-17}$ |
| Paramythia montium | 180,000 | 40,000 | 160,000 | 340,000 | 700,000 | $4 \times 10^{-6}$ | $2 \times 10^{-12}$ | $7 \times 10^{-11}$ | $8 \times 10^{-11}$ | $3 \times 10^{-11}$ | $1 \times 10^{-10}$ |
| Peneothello sigillata | 150,000 | 50,000 | 20,000 | 240,000 | 1,800,000 | $8 \times 10^{-6}$ | $2 \times 10^{-18}$ | $6 \times 10^{-11}$ | $3 \times 10^{-6}$ | $8 \times 10^{-14}$ | $4 \times 10^{-6}$ |
| Sericornis nouhuysi | 390,000 | 120,000 | 120,000 | 130,000 | 1,500,000 | $4 \times 10^{-11}$ | $3 \times 10^{-11}$ | $1 \times 10^{-13}$ | $3 \times 10^{-6}$ | $8 \times 10^{-11}$ | $2 \times 10^{-10}$ |
| Aethomyias papuensis | 80,000 | 90,000 | 130,000 | 200,000 | 1,400,000 | $9 \times 10^{-11}$ | $6 \times 10^{-6}$ | $1 \times 10^{-13}$ | $2 \times 10^{-5}$ | $5 \times 10^{-6}$ | $3 \times 10^{-16}$ |

| | Nm Wilhelm to Huon | Nm Huon to Wilhelm | Nm Wilhelm to Scratchley | Nm Scratchley to Wilhelm | Nm Huon to Scratchley | Nm Scratchly to Huon |
|---|---|---|---|---|---|---|
| Pachycephala schlegelii | 5.1 | 0.0 | 0.0 | 0.0 | 0.6 | 0.0 |
| Paramythia montium | 0.7 | 0.0 | 0.0 | 0.0 | 0.0 | 0.0 |
| Peneothello sigillata | 1.2 | 0.0 | 0.0 | 0.1 | 0.0 | 0.1 |
| Sericornis nouhuysi | 0.0 | 0.0 | 0.0 | 0.4 | 0.0 | 0.0 |
| Aethomyias papuensis | 0.0 | 0.5 | 0.0 | 2.6 | 0.5 | 0.0 |

for the montane species in New Guinea are substantially lower (0–1.8 dispersing individuals per generation between populations), but dispersal is nonetheless evident; genetically well-separated highland populations are clearly also able to cross significant barriers, albeit at a reduced rate. This contrasts with findings from South America, where rivers often represent impenetrable barriers to dispersal and gene flow both for lowland and highland species[5,39].

**What influence have dramatic Pleistocene climate fluctuations had on population demographics?** Demographic analyses using fastsimcoal2 and PSMC suggest that all species included in our study have experienced marked demographic changes in their past history. The most dramatic oscillations occurred during the Pleistocene (2.6 Mya to 11.7 Kya); this was preceded by a long period of relative demographic stability during the Pliocene and Miocene.

PSMC curves of populations of the same species on Buru and Seram show remarkable similarity (Fig. 2 except *Ceyx lepidus*), especially considering the significant genetic differentiation between them ($F_{ST} = 0.09$–$0.16$). This strongly indicates that shared regional climatic and environmental fluctuations, rather than island-specific community dynamics or stochasticity, have governed population sizes in the Moluccan islands during the past few million years.

In New Guinea, the demographic histories (PSMC curves) are highly similar for many species that are members of different families (Figs. 3 and 4), also suggesting that the environmental conditions rather than lineage characteristics have influenced population sizes through time. The majority of species in New Guinea have undergone significant increases in population sizes during the early to mid-Pleistocene, followed by equally significant decreases in the late Pleistocene. These marked demographic oscillations coincide with a time when climate fluctuations were most dramatic. On very recent time scales, PSMC is known to be less accurate, and we therefore rely more on analyses using fastsimcoal2 to infer demographic histories during the recent past (Table 1). Fastsimcoal2 analyses suggest low-to-moderate population sizes that never reach the magnitudes seen in the mid-Pleistocene.

It is well-known that climatic fluctuations in the Pleistocene have strongly influenced the demographic histories and distributions of species[40,41]. Many species underwent range shifts as the extent of suitable habitats changed, and it is hypothesized that montane species in the tropics may have descended to lowland regions during colder periods. Such elevational range shifts are also likely to affect population densities as more suitable habitat space becomes available[41].

While in general tropical regions were not as heavily affected as regions at higher latitudes, the effect of the Pleistocene climatic changes in the Indo-Pacific was significant, and is believed to have caused the extinction of many large vertebrate taxa, while a range of medium- and small-bodied species underwent major geographic range shifts[42]. Hence, we suggest that the population size oscillations observed herein reflect the severe Pleistocene climatic oscillations[43–45], in combination with the marked tectonic rearrangements and uplift that have greatly affected the region[16,17].

For the New Guinean species, we find strikingly contrasting demographic patterns between populations from Mount Wilhelm and from Huon in the more recent past (Fig. 2). In general, the Mount Wilhelm and Huon populations showed identical demographic trends until the mid-Pleistocene, at which time, demographic patterns become more idiosyncratic. For the majority of the montane species, there are clear population divergences in PSMC curves that are easily interpretable as splits between Central Range (Mount Wilhelm and Mount Scratchley)

and Huon populations, with the majority of these splits supported by both the PSMC and fastsimcoal2 analyses. Generally, the Huon population sizes decreased, while Mount Wilhelm populations remained relatively stable until the present. This pattern is evident for the majority of montane taxon pairs, but not for the two lowland species, reflecting the comparatively more stable climatic conditions that high elevation taxa are exposed to as they track their preferred niche up and downslope during climatic oscillations. Both sets of demographic analyses (PSMC and fastsimcoal2) suggest that effective population sizes are higher on Mount Wilhelm than in Huon, as would be expected given the more limited geographic extent in the latter.

**Does adaptation to environmental extremes along elevational gradients lead to ecogeographic isolation and parapatric speciation?** In contrast to the distinct genetic differentiation found between most montane taxon pairs, our results show little evidence of genetic differentiation along individual elevational gradients (Supplementary Fig. 3A, B), with the exception of *Ifrita kowaldi* for which individuals at 2300 and 2900 m a.s.l. show some genetic divergence ($F_{ST} = 0.03$, Supplementary Fig. 3B and Supplementary Table 2). While *I. kowaldi*, may represent a case of incipient parapatric speciation along an elevational gradient, other species sampled in our study appear to have diverged exclusively in allopatry. Most species are likely to have inhabited Seram, Buru or New Guinea's Central Range for more than 1 My, leaving ample time for incipient speciation to occur. Yet, the levels of differentiation across elevations within the same gradient are negligible, when generalizing across the whole genome. This suggests that parapatric speciation along elevational gradients does not appear to be an important mechanism for the formation of montane avian diversity[10,46]. For example, in *Paramythia montium*, samples from Mount Wilhelm, Mount Scratchley and Huon show high genetic differentiation among populations, but comparisons of samples from different elevations along Mount Wilhelm suggest genetic differences to be virtually zero. Even in the most extreme cases, *Origma robusta* and *Rhipidura albolimbata*, in which we compared individuals sampled at elevational sites 2000 m apart, no substantial genetic differences were found. However, recent work has shown that divergence with gene flow may not manifest as a genome-wide phenomenon. Instead, it may proceed via selection acting to create small 'islands of differentiation' within the genome against a background of negligible differentiation[22]. Thus, while our analyses summarizing across the whole genome do not find support for parapatric speciation along elevational gradients, more sophisticated 'islands of differentiation' analyses might reveal a different story.

**Montane dispersal, rather than recruitment from the lowlands account for the current distribution of montane taxa in New Guinea.** Taxon pairs at higher elevations in New Guinea are significantly more genetically differentiated relative to taxon pairs at lower elevations. This pattern could suggest that populations are recruited from the lowlands and then progressively become more isolated from one another as they move into the highlands. However, our demographic analyses (fastsimcoal2 and PSMC) suggest that the two most diverged taxon pairs, *I. kowaldi* and *P. montium*, started diverging less than 100 Kya, which significantly postdates the species' formation and their probable adaptation to highland environments. Thus, it appears more plausible that these and other mountain species were already montane 1 Mya, and that the diverging population-level PSMC trajectories reflect more recent mountain to mountain dispersal and colonization. Further, if populations were recruited from the lowlands, this would leave a signature of declining population sizes (Ne)

through time, and relatively small population sizes at present (reflecting the more limited available area). We do not find such patterns. Finally, under the lowland recruitment hypothesis, isolated populations at high elevations are expected to be highly sedentary. Fastsimcoal2 suggests low but non-negligible rates of migration for even the most highly diverged species in our study (in the order of $10^{-7}$ migrants per generation), implying that montane populations do, in fact, have the ability to disperse.

Instead, our results emphasize the relative importance of dispersal between montane regions. Further, the patterns of dispersal that we recover in our demographic data support Diamond's hypothesis that these dispersal events follow the logic of island biogeography theory. Because of the physical configuration of New Guinea's mountains, with a dominating Central Range akin to a "continent" and several small outlying ranges that can be viewed as "islands", Diamond[47] suggested that montane distributions are driven by repeated colonization of the outlying ranges from the Central Range. The smaller populations of the outlying ranges have a higher probability of going extinct, so in the case considered here, colonization patterns would be predicted to become uni-directional from the Central Range to Huon. Under this hypothesis, highland species continuously colonize other highland regions via direct dispersal from mountaintop to mountaintop.

While our data clearly demonstrate that population sizes of all species analyzed herein have fluctuated markedly due to climatic and geological dynamics, several lines of evidence indicate that the divergent patterns in demographic (PSMC) trajectories between populations of the Central Range and Huon populations are best interpreted as reflecting a net colonization of Huon from the Central Range. All favored models in the demographic analyses in fastsimcoal2 included migration, which suggests continuous gene flow between Mount Wilhelm and Huon (and also between these two and Mount Scratchley), such that highland species are able to disperse between separated mountain regions. While dispersal rates are often similar in both directions between Huon and the Central Range (Table 1), the smaller and more fluctuating populations in the Huon mountains are likely more prone to go extinct, thereby opening up available ecological space for new colonizers from the Central Range.

Following dispersal, newly formed Huon populations separate from those of Mount Wilhelm, with population sizes (Ne) decreasing in these "founder" populations. Further support for this dispersal scenario is a visible increase in population sizes (Ne) for the majority of species preceding the split of demographic histories (PSMC curves). This increase either reflects the actual range expansion event prior to divergence, or a build-up of individuals (i.e. potential dispersers) within the source range. While the Huon populations decrease due to the population bottleneck associated with a founder event and reduced inhabitable area, Central Range populations remain stable or increase after the split. If we assumed a lowland origin followed by upslope range contraction into separate mountain ranges and/or vicariant speciation, we would expect that Central Range populations would also decline post-divergence.

Montane avifaunas might have had greater opportunity to disperse between their preferred habitat during glacial maxima, when elevational habitat belts shifted downslope and possibly connected previous isolated montane populations. However, many species in our study show appreciable divergences that significantly predate recent glacial maxima. Moreover, speciation via habitat connectivity and divergence is a vicariant process, and the demographic histories (PSMC curves) do not reflect range contractions following population divergence. This mechanism is not, however, mutually exclusive with direct mountain-to-mountain dispersal. If gaps between tracts of montane forest

shrank during glacial maxima, this would have reduced distances between allopatric populations, and increased the probability of individuals dispersing between them. Consequently, dramatic climatic oscillations during the Pleistocene would favor both potential mechanisms.

Our comparisons using whole genomic data of population pairs of eighteen Indo-Pacific bird species shed new light on the formation of montane diversity across tropical island settings. The genomic data generated in this study allow us to infer the underlying mechanisms leading to the formation of highland taxonomic diversity, revealing that levels of genetic differentiation varies predictably across taxa as a result of species distributions and ecology. Specifically, we find that (i) "hard" oceanic barriers generate greater differentiation than "soft" lowland barriers, (ii) a higher altitudinal floor is associated with greater differentiation between populations, (iii) physical barriers generate greater differentiation than environmental gradients and (iv) demographic fluctuations are greater in smaller mountain ranges than larger ones. These data further highlight the importance of dramatic Pleistocene climatic oscillations, which have influenced the extent of montane forests, to determine temporal patterns of population expansion and contraction along the elevational gradients. Intermittent dispersal between disjunct New Guinean montane regions is repeatedly documented across taxon pairs and is likely an important factor underlying the expansion and build-up of montane passerine bird diversity in this area. Dispersal is much less frequent between taxon pairs in the Moluccan islands. While it is often assumed that islands and mountaintops represent evolutionary dead ends, we show that disjunct montane populations of the same species can maintain gene flow, as well as their capacity to further colonize new regions. This has implications for how we view the formation of montane biotas, and how we approach the conservation of its present diversity.

## Methods

This research complies with all relevant ethical regulations in Indonesia and in Papua New Guinea. Research in New Guinea was approved by the State Ministry of Research and Technology (RISTEK, permits SURAT IZIN PENELITIAN: 013/SIP/ FRP/I/2011 and 026/SIP/FRP/I/2012). The PNG National Museum and art gallery and the Conservation and Environment Protection Authority (CEPA) of Papua New Guinea provided research permits and export permits. Research in Indonesia was approved by the State Ministry of Research and Technology (RISTEK, permits SURAT IZIN PENELITIAN: 013/SIP/FRP/I/2011 and 026/SIP/FRP/I/2012), the Ministry of Forestry (Republic of Indonesia), the Research Center for Biology, Indonesian Institute of Sciences (RCB-LIPI) and the Bogor Zoological Museum. Collectively, these institutions provided permits to carry out fieldwork in Indonesia and to export select samples.

**Sampling**. Our study included populations pairs of eighteen bird species; one non-passerine species (Coraciiformes) the Moluccan dwarf kingfisher (*Ceyx lepidus*) and seventeen passerine birds (Passeriformes) representing the three superfamilies (Meliphagides [6 species], Corvides [5 species] and Passerides [6 species]) (Fig. 1; Supplementary Fig. 5; Supplementary Table 3 and Supplementary Data 2). For thirteen species (*Pachycephala schlegelii, Sericornis nouhuysi, Aethomyias papuensis, Melipotes fumigatus/ater, Peneothello sigillata, Ptiloprora guisei, Rhipidura albolimbata, Origma robusta, Paramythia montium, Melanocharis versteri, Ifrita kowaldi, Toxorhamphus novaeguineae* and *Melilestes megarhynchus*), samples were obtained along elevational transects in two separate mountain ranges on New Guinea: the northeast slope of Mount Wilhelm (4509 m a.s.l.) in the Central Range and in the Saruwaged Range (4121 m a.s.l.) in the Huon Peninsula (hereafter "Huon"). Eleven of the thirteen species were obtained from medium or high elevations, and specimens were collected at different sampling stations between 1700 and 3700 m a.s.l. Two lowland species, *T. novaeguineae* and *M. megarhynchus*, were included, with samples obtained between 50 and 1200 m a.s.l. Because the widespread montane species *M. fumigatus* is not present in the Saruwaged Range, we compared populations of this species from Mount Wilhelm with the congeneric *M. ater*, which is endemic to the Huon Peninsula. As *P. guisei*, is replaced at higher elevations by its congener *P. perstriata*, we included samples of *P. perstriata* from Mount Wilhelm and Mount Scratchley (*P. perstriata* is not present in the Huon Peninsula) to assess the potential for admixture between these two species. The Central Range and the Saruwaged Range are separated by the comparatively arid Markham/Ramu River valley, which features extensive river floodplains, *Imperata*

*cylindrica* grassland, and anthropogenic gardens. To assess the connectivity of populations of lowland species, we obtained samples from a third population from appropriate habitats in the biogeographic transition zone between Mount Wilhelm and Huon, north of the Markham/Ramu River. As the nearest forested corridor between the Central and Saruwaged ranges, these samples allow us to directly gauge the strength of the river barrier. For a subset of five of the New Guinean highland species (*P. sigillata*, *P. montium*, *P. schlegelii*, *S. nouhuysi* and *A. papuensis*), we additionally sampled individuals from a third population from Mount Scratchley, which is a continuation of the Central Range in south-eastern New Guinea. By doing so, we are able to contrast population demographic trajectories between mountains within the Central Range as well as our primary contrast with the well-isolated outlying mountains of the Huon Peninsula.

Five species in our study (*Ceyx lepidus*, *Pachycephala macrorhyncha*, *Ficedula hyperythra*, *Ficedula buruensis* and *Thapsinillas affinis*) were obtained from the Indonesian islands of Buru and Seram, within the Moluccan Islands to the west of New Guinea. Buru is 9505 km² with a maximum elevation of 2428 m a.s.l., while Seram is 17,100 km² with a maximum elevation of 3027 m a.s.l. The two oceanic islands are separated by ~75 km of ocean, but due to their close proximity share a substantial amount of bird species. On Buru, samples were collected in the northwest part of the island at elevations ranging between 227 and 1425 m a.s.l., while on Seram samples were obtained from the north coast and inland between elevations of 185–1300 m a.s.l. (see Supplementary Data 2 for detailed locality information).

**Genome sequencing and de novo assembly.** To facilitate demographic analyses, a de novo genome was assembled for fifteen of the eighteen species using the 10X Chromium linked-reads platform at the Science for Life Laboratory in Stockholm, Sweden (SciLifeLab; Supplementary Table 3). For the remaining three species (*Aethomyas papuensis*, *Pachycephala macrorhyncha* and *Ficedula buruensis*), no de novo genome was assembled since the study already included the de novo genome of another closely related species (*Sericornis nouhuysi*, *Pachycephala schlegelii* and *Ficedula hyperythra*, respectively; note that *A. papuensis* was formerly placed in the genus *Sericornis* until a recent revision placed it in the genus *Aethomyias* within the same subfamily[48]).

Total DNA from muscle samples (stored at −80 °C) was extracted using a Thermo Scientific KingFisher Duo magnetic particle processor (ThermoFisher Scientific) with the KingFisher Cell and Tissue DNA Kit. Only one individual was used as a template for each genome sequence. Following quantification on a Qubit Fluorometer (ThermoFisher Scientific) and validation of the extract quality on an agarose gel, the extracts were used to prepare Illumina sequencing libraries. Libraries were sequenced on an Illumina HiSeqX (HiSeq Control Software HD 3.5.0.7/RTA 2.7.7) with a 2 × 151 bp setup using HiSeq X SBS chemistry. The Bcl to FastQ conversion was performed using BCL2FASTQ v.2.19.1.403 from the CASAVA software suite.

All genomes were assembled de novo using Supernova v2.0.1[49] as incorporated in the nextflow-core Neutronstar workflow[50]. Standard assembly statistics were obtained including total assembly length, number of scaffolds, scaffold N50 and percentage of base assembly missing in scaffolds larger than 10 kb. These statistics give an idea of the contiguity of the assembly. To validate the assembly, independent contiguity metrics were also obtained using QUAST v5.0.1[51], a quality assessment tool for genome assemblies. In addition, BUSCO v4.0.2[52] was used to assess the assembly quality in terms of conserved gene recovery. The software inspects the de novo assemblies searching for Benchmarking Universal Single-Copy orthologs called BUSCOs. A set of 300 eukaryote orthologs was used. BUSCOs were classified as complete and single-copy, complete and duplicated, fragmented or missing. All de novo assembled genomes are deposited on NCBI under SRA accession: PRJNA 637995.

**Resequencing data.** A total of 226 individuals representing the eighteen species were re-sequenced at SciLifeLab, Stockholm (Supplementary Data 2). The number of individuals sequenced per species ranged from 9 in *Thapsinillas affinis* to 18 in *Ptiloprora guisei* and *Pachycephala macrorhyncha*. Genomic DNA was extracted from either muscle tissue or blood samples stored at −80 °C using a Thermo Scientific KingFisher Duo magnetic particle processor (ThermoFisher Scientific) with the KingFisher Cell and Tissue DNA Kit. DNA was quantified using a Qubit Fluorometer (ThermoFisher Scientific). Libraries were made using either Illumina TruSeq DNA Library Preparation Kits (high DNA input) or Rubicon Thruplex DNA-seq Kits (low DNA input), with an average fragment size of 350 bp in both cases. Samples were sequenced on an Illumina HiSeqX (HiSeq Control Software HD 3.5.07/RTA v2.7.7) with a 2 × 151 bp setup using HiSeq X SBS chemistry. The Bcl to FastQ conversion was performed using BCL2FASTQ v.2.19.1.403 from the CASAVA software suite. All resequencing data are deposited on NCBI under SRA accession: PRJNA580350.

**Resequencing data filtering and SNP discovery.** Data quality of the Illumina reads was initially assessed using FastQC (http://www.bioinformatics.babraham.ac.uk/projects/fastqc) and the FASTX-Toolkit (http://hannonlab.cshl.edu/fastx-toolkit), after which we trimmed the first 5 base pairs (bp) and the last 11 bp from the forward reads, and the first 5 bp and the last 21 bp of the reverse reads. Next, adapters and low

quality bases were removed with Trimmomatic v0.36[53], using a minimum Phred score of 15. Subsequently, paired and unpaired reads were aligned separately to the respective de novo genomes using BWA v0.7.1[54]. Aligned data were pre-processed using Picard v2.6.0[55], which included merging (MergeSamFiles), coordinate-sorting (SortSam) and identifying duplicate reads that may arise during library preparation (MarkDuplicates).

Subsequently, a vcf file with all SNP variants per species was constructed with the mpileup and call commands in BCFtools v1.9[56]. We excluded sites for which read depth was less than 10 or more than 100, sites for which quality was below 20 and sites in proximity to indels. As a final step, SNP variants were filtered using vcftools v0.1.14[57] so that only biallelic SNPs were retained and only loci present in all individuals sequenced in each species pair were processed.

**Population structure analysis.** All population structure analyses were conducted using reduced SNP datasets after linkage disequilibrium-based variant pruning implemented in PLINK v1.90b3.42[58]. The indep-pairwise option was used with a window size in the variant count of 100, a variant count of 5 to switch window at the end of each step and a r² threshold of 0.5. Standardized genetic differentiation statistics between population pairs were calculated using vcftools v0.1.14[57] in accordance with Weir and Cockerham[59]. When populations included more than one individual for two elevations, differences between samples from different elevations were also computed.

Standardized pairwise $F_{ST}$ values were used to conduct a Principal Component Analysis (PCA) in order to visualize population structure using the princomp function in R[60]. Prior to the analysis, the PCAngsd software v0.95[61] was used to remove related individuals based on a kinship matrix. Individuals with a pairwise kinship coefficient higher than 0.06 were considered to be closely related.

Population structure was further investigated using the Bayesian assignment approach implemented in STRUCTURE[62], a model-based clustering algorithm that infers the most likely number of groups (K) in the data. The analysis was performed with K = 1–6, assuming an admixture model, correlated allele frequencies and without population priors. In order to choose appropriate burn-in and run lengths several test runs were run at different Ks and different lengths (from $10^5$ to $10^6$) and checked for consistent estimates of the parameters P, Q and lnPr(X | K). Ultimately, a burn-in of 10,000 steps followed by 100,000 additional Markov Chain Monte Carlo iterations were performed. For each K, 10 independent runs were conducted to check the consistency of the results. The most likely K was inferred using the method of Evanno et al.[63], which measures the steepest increase of the ad hoc statistic ΔK based on the rate of change in the log probability of data between successive K values.

**Regression analysis.** We estimated the relationship between genetic differentiation ($F_{ST}$) values and elevational floor using a Spearman's rank correlation. If lineages colonize mountains from the lowlands, followed by range contraction and differentiation in the highlands, we would expect a strong correlation between $F_{ST}$ and elevational floor (the lower trailing edge of the distribution) coupled with no dispersal between populations. We obtained elevational information for New Guinea species (Fig. 1) from Beehler and Pratt[38] supplemented by information from Sam and Koane[64] as well as from our own fieldwork in Papua New Guinea[65].

**Demographic history inference.** Coalescent methods can be applied to infer past population dynamics and demographic changes over historical time scales and to estimate current and historical population sizes with high resolution. We investigated the demographic history of the bird species in our study using two complementary approaches that perform well at different time scales: PSMC and fastsimcoal2. In both programs, results were scaled using a generation time of 2 years and a mutation rate of $3 \times 10^{-9}$ per nucleotide per generation for all passerine species and a rate of $2 \times 10^{-9}$ per nucleotide per generation for *Ceyx lepidus* based on the rates reported for passerine birds and carmine bee-eater *Merops nubicus*[66].

**Demographic analyses using PSMC.** For each de novo sequenced individual (one per species) and all re-sequenced individuals, a consensus sequence was generated using BCFtools v1.9[56], following the same approach as Nadachowska-Bryzska et al.[66]. First, the raw Illumina reads were trimmed using FASTX-Toolkit v0.0.14 (http://hannonlab.cshl.edu/fastx-toolkit). After excluding non-autosomal regions, variants were called with the mpile and call commands in BCFtools. The following filtering criteria were applied: we excluded sites for which read depth was less than 10 or more than 100; sites with Phred quality scores below 20; and sites in the proximity of indels. Subsequently, we used the 'consensus' command in BCFtools to incorporate all variants into a single sequence using IUPAC codes.

The consensus sequence was then divided into non-overlapping 10 bp bins, which were scored as heterozygous if there was at least one heterozygote nucleotide position in the bin, and otherwise scored as homozygous. The total number of iterations was set to 25; T max (-t) was set to 15; the initial mutation/recombination ratio (-r) was set to 5; and the atomic time interval pattern (-p) was set to "4 + 25*2 + 4 + 6". Bootstrapping was performed by splitting the data into shorter segments that were then randomly sampled with the replacement for a total of 100 rounds.

**Demographic analyses using fastsimcoal2.** In addition to PSMC, we also modeled demographic history with an approach based on the allele frequency spectrum (AFS). We used the program fastsimcoal2[25] to generate the expected AFS under different models of historical divergence using coalescent simulations, select the best-fit model using the composite multinomial likelihood, and infer demographic parameters. We were particularly interested in inferring current and ancestral effective population size, time of divergence, the occurrence of historical events allowing for population size changes (bottlenecks, founder effects or sudden expansion events) and the presence of gene flow and its directionality (symmetric/asymmetric). To this end, we maximized the likelihood of the observed AFS under a total of 8 different demographic scenarios applied to each population pair across all species. The simplest model was (i) a null model with no growth and no migration between populations in isolated mountain ranges. We also tested (ii) a model with migration but no growth, (iii) a model with exponential growth but no migration and (iv) a model with both exponential growth and migration. Furthermore, we tested the likelihood of a demographic event in the past that led to the occurrence of a bottleneck both (v) with and (vi) without migration, and an event leading to a sudden expansion, again (vii) with and (viii) without migration. For those species in which the PSMC plots showed a marked difference in the demographic history of the two populations we compared, we additionally tested (ix) a model with a bottleneck in the first population and a sudden expansion in the second population and the reversed scenario, i.e. (x) a model with a sudden expansion in the first population and a bottleneck in the second population.

The AFS for all populations were directly generated from the final variant calls (vcf files) using the scripts in https://github.com/shenglin-liu/vcf2sfs. Exact models and priors for all parameters are described in Supplementary Methods. For every demographic model, 100 independent estimations with different initial parameter values were run and results for the estimation with the highest likelihood were reported. The best-fitting demographic model was identified using Akaike's Information Criterion (AIC) score[67].

**Reporting summary.** Further information on research design is available in the Nature Research Reporting Summary linked to this article.

## Data availability

All de novo assembled genomes are deposited on NCBI under SRA accession: PRJNA 637995. All resequencing data are deposited on NCBI under SRA accession: PRJNA580350.

## Code availability

Code for the fastsimcoal2 demographic models is provided in Supplementary Methods in the Supplementary Information

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

## Acknowledgements

K.A.J., A.H.R., P.Z.M. and J.D.K. thank all the staff and field assistants that facilitated fieldwork in Papua New Guinea. Notably, the Binatang Research Center and local communities along the Mount Wilhelm gradient, as well as people from the villages of Kanga, Keglsugl, Towet, Yawan, and Wanang. We are also grateful for the assistance provided by the PNG National Museum and art gallery and the Conservation and Environment Protection Authority (CEPA) of Papua New Guinea for research permits and export permits. Furthermore, we thank the State Ministry of Research and Technology (RISTEK, permits [SURAT IZIN PENELITIAN: 013/SIP/FRP/I/2011 and 026/SIP/FRP/I/2012 to K.A.J.]); the Ministry of Forestry, Republic of Indonesia; the Research Center for Biology, Indonesian Institute of Sciences (RCB-LIPI); and the Bogor Zoological Museum for providing permits to carry out fieldwork in Indonesia and to export select samples. We acknowledge support from the National Genomics Infrastructure in Stockholm funded by Science for Life Laboratory, the Knut and Alice Wallenberg Foundation and the Swedish Research Council, and SNIC/Uppsala Multidisciplinary Center for Advanced Computational Science for assistance with massively parallel sequencing and access to the UPPMAX computational infrastructure. J.D.K. was supported by an Individual Fellowship from Marie Sklodowska-Curie actions (MSCA-792534). K.S. acknowledges ERC grant BABE 805189. K.A.J. acknowledges a National Geographic Research and Exploration Grant (8853-10), a Carlsberg Foundation Expedition Grant (CF15-0079), the Dybron Hoffs Foundation and the Corrit Foundation for financial support for fieldwork in Indonesia and Papua New Guinea. K.A.J., A.H.R. and J.M.P. are most grateful for the financial support received from the Villum Foundation (Young Investigator Programme, project no. 15560).

## Author contributions

K.A.J. conceived the study. A.H.R., J.D.K., P.Z.M., B.G.F., K.S., T.H., B.I., B.K., G.M., L.P. and K.A.J. carried out the fieldwork. J.M.P. and M.I. performed the laboratory work. J.M.P., M.P.K.B., J.D.K. and T.S.K. carried out the analyses. K.A.J. wrote the manuscript with input from all authors

## Competing interests

The authors declare no competing interests.
