## [Peer Review File · Nature Communications]

Peer Review comments, first round review –

Reviewer #1 (Remarks to the Author):

This submission uses whole-genome data to compare patterns of differentiation across pairs of montane land bird species that are separated by different kinds of historical barriers, including 'hard' barriers (permanent ocean) and 'soft' barriers (lowland habitat that might be more permeable). Although there is much to like about the details of these explorations and what they have to say about the histories of these particular communities and species, I feel that this paper substantially over-reaches in trying to answer large questions that are beyond the scope of these data to address. I will therefore focus my review on these most overarching limitations, while underlining that I do think these data have utility for some of the more focused aspects of this study.

This paper joins a long history of trying to pull integrative signal out of the comparative biogeography of taxa with complex histories. I would argue that one of the lessons learned over the 40+ years of this research history is that the historical idiosyncrasies of the taxa in these systems is almost always higher than one would expect at the outset. For example, this is evident in the current study, where one of the *Ficedula* has a distinctly different history than the other island taxa included in these comparisons. This general fact of taxon-specific historical contingency has led in this field to two practical truisms. The first is that it is risky to try to pull reliable signal from point comparisons (such as two mountains or two islands, as in this study) while not also incorporating a reconstruction of differentiation across the full range of each taxon. The second is that owing to this variance across the histories of the different taxa, really large sample sizes are required to make generalizable conclusions. This is very challenging to accomplish, but it has been done (for example, in studies of dispersal ability and differentiation levels in South American birds). Here, the comparison of just one mountain gap with just one island gap (each with its own set of non-comparable histories), and the relatively modest number of taxa compared for each, limits the power of this study to identify generalizable patterns.

In first reading the abstract of this paper, I was most drawn to its potential to address the possibility of in situ montane speciation via selection across ecological gradients. Some of the authors of the present study have impressive histories of work on this somewhat controversial topic. Unfortunately, I think they are making a basic mistake in how they analyze their elevational data along these gradients. There is a vast recent literature on these processes showing that divergence-with-gene-flow does not manifest as a genome-wide phenomenon (at least, not until the taxa are so differentiated that gene flow has ceased). Instead, it proceeds via selection acting to create small 'islands of differentiation' within the genome against a background of negligible differentiation. Therefore, even if ecological gradients were driving in situ divergence in these montane birds, one would not expect to even glimpse it in the kinds of gross whole-genome F_{st} or STRUCTURE comparisons that this paper presents. For other reasons, I agree with the authors that this kind of in situ montane speciation is likely to be rare (at best), but I do not think that their gross differentiation approach is a viable method for testing this hypothesis. As an aside, the sample sizes for these within-taxon elevational comparisons are also not sufficient.

I caution the authors for potentially reading too much into the fastsimcoal2 plots of demographic parameters. fastsimcoal has been a workhorse for this kind of inference for almost a decade, partly because there are few good analytical alternatives, but these models have real limitations. What are the error estimates on these parameters? An entry point into this topic is the 2017 book chapter by Salmona et al. *Inferring Demographic History Using Genomic Data*.

Reviewer #2 (Remarks to the Author):

I enjoyed reading the paper and found it to be another useful contribution of molecular population genomics and demographics to understanding the dynamics of montane bird evolution in the Indo-

Pacific. Along with paper like the recent Garg et al paper in BMC E and E, this is developing into a model suite of papers and I thank and congratulate the authors.

Most of my comments are on the attached pdf. Some key issues are:

I found the aims a little confused in so far as selection and adaptation seemed to come and go in various parts of the paper. Perhaps that could be clarified.

The paper is a little lop-sided in its discussion of the New Guinean results relative to the Moluccan results. That is understandable as the story seems more complicated in New Guinea. Yet after I had finished reading the paper, I went back into the earlier sections looking for some more details about the Moluccas and New Guinea together. In particular, the ages of the mountains and the sea/land gaps in the Moluccas and New Guinea need to be clearer and easier to compare. Are the Moluccan results just being ascribed (except in *Ficedula* I think) to vicariance either side of the ocean gap? Is that why New Guinea needs more detailed discussion? Maybe lead the reader a little more kindly through that.

Patton and Smith 1992 as described in the comments on the pdf is worth a look I suspect.

Lots of long sentences. Break them up more, please! I have suggested some ways to tackle this.

Reviewer #3 (Remarks to the Author):

This paper takes advantage of a favorable geographic situation to address many basic debated questions of evolution and biogeography: questions of speciation, genetic differentiation, dispersal, taxon cycles, population sizes, and others.

The geographic situation is the mountains of New Guinea and of the Moluccan islands, where the authors measured and compared genetic differentiation across two different barriers: the lowland barrier between New Guinea's Central Range and its Huon Mountains, and the ocean barrier between Moluccan islands.

At the outset of this review, I'll mention a strength of this study that the authors modestly never mention and that is likely to make it unique: the extreme physical, logistic, and permitting obstacles to doing research in the mountains of Buru, Seram, New Guinea's Central Range, and New Guinea's Huon Mts. To accomplish this study in just one of those four mountains would have been difficult; to accomplish it in all four mountains, difficult to the extreme.

The only significant problem that I find in this important paper is to present clearly the diverse sets of results. Hence my review will consist of my summary of the results, followed by my suggestions for presentation.

SUMMARY OF RESULTS. The authors address nine questions, and reach nine sets of conclusions:

1. Genetic differentiation of mountain birds is generally greater across the ocean barrier than across the lowland barrier.
2. Within this generalization, there are some differences among species within the same fauna.
3. Differentiation is greater between populations of montane species than between populations of lowland species.
4. Differentiation is greater for montane species with higher altitudinal floors, in New Guinea but not in the Moluccas.
5. There were marked fluctuations in population sizes during the Pleistocene.
6. There is negligible differentiation along elevational gradients, hence no evidence for sympatric speciation in this study.

7. The montane populations studied originated by jumping between mountains. There is only limited support for origins by push-pull, as in taxon cycles.

8. The direction of colonization is predominantly from a larger mountain range to a smaller mountain range.

9. Dispersal can be estimated from genetic evidence. I consider this finding a big deal about which the authors are being too modest. Knowledge of dispersal is fundamental to understanding biogeography. We know a lot about bird dispersal in Europe and North America, from the observations of millions of bird-banders. Such information is almost non-existent for New Guinea. This paper's dispersal estimates from genetic evidence are currently unique for the New Guinea region.

SUGGESTIONS FOR FRAMING THE PRESENTATION OF THESE NINE QUESTIONS AND CONCLUSIONS MORE CLEARLY.

In the first paragraph of the introduction, succinctly mention several of these major detailed questions,, as providing the motivation for this paper.

In the second paragraph of the introduction, explain your favorable test situation for answering these questions: comparison of the ocean barrier between Buru and Seram with the lowland barrier between New Guinea's Central Range and Huon Mts., as those barriers affect genetic differentiation of 18 montane forest bird species.

In the last paragraph of the introduction, succinctly list all nine questions.

In the first paragraph of the discussion, orient readers by succinctly listing (one sentence each) your conclusions about these nine questions, before plunging into details of your conclusions. For example, you could modify my draft "summary of results" above.

OTHER SUGGESTIONS FOR IMPROVING CLARITY

Some important concepts or acronyms aren't defined at the first mention in the text. Please define them: parapatric speciation in situ (line 130), FST (line 183), STRUCTURE (line 187), $K = 1,2,3,\dots$ (line 213).

In addition to those definitions in the text at first mention, gather and repeat all definitions of acronyms in a footnote near the beginning of the paper.

SPECIFIC SUGGESTIONS

Line 243: the expression "minimum elevation" is unclear. Say instead "altitudinal floor," and add a phrase to explain.

Line 135: the two New Guinea mountain ranges are not separated by a river barrier as stated in this line, but instead by a lowland barrier. The Ramu and Markham Rivers themselves constitute only a small fraction of this lowland barrier's width.

Line 216: say that Mt. Scratchley is in the Central Range as is Mt. Wilhelm, but 400 kilometers to the east, and separated by a much weaker lowland barrier than that between Mt. Wilhelm and the Huon Mts. Otherwise, readers won't understand the significance of Mt. Scratchley: it's not enough for readers to guess the significance from the figured map.

Lines 437 - 470, section on jumping. This section argues that montane species colonize other mountains by jumping, not by push-pull. That may be true, but mention that the ORIGIN of a montane species is a separate question; that question of the origin of montane species is the question addressed by explanations involving taxon cycles and push-pull.

REVIEWER COMMENTS

Reviewer #1 (Remarks to the Author):

This submission uses whole-genome data to compare patterns of differentiation across pairs of montane land bird species that are separated by different kinds of historical barriers, including 'hard' barriers (permanent ocean) and 'soft' barriers (lowland habitat that might be more permeable). Although there is much to like about the details of these explorations and what they have to say about the histories of these particular communities and species, I feel that this paper substantially over-reaches in trying to answer large questions that are beyond the scope of these data to address. I will therefore focus my review on these most overarching limitations, while underlining that I do think these data have utility for some of the more focused aspects of this study. This paper joins a long history of trying to pull integrative signal out of the comparative biogeography of taxa with complex histories. I would argue that one of the lessons learned over the 40+ years of this research history is that the historical idiosyncrasies of the taxa in these systems is almost always higher than one would expect at the outset. For example, this is evident in the current study, where one of the *Ficedula* has a distinctly different history than the other island taxa included in these comparisons. This general fact of taxon-specific historical contingency has led in this field to two practical truisms. The first is that it is risky to try to pull reliable signal from point comparisons (such as two mountains or two islands, as in this study) while not also incorporating a reconstruction of differentiation across the full range of each taxon.

RESPONSE: These are a valid concerns and we do agree with much of what the reviewer states. However, the specific aim in this paper is to evaluate how two populations evolve on either side of a barrier and the study was set up this way from the very beginning, and we believe that our data are still an effective way to do this. The *Ficedula* example highlighted by the reviewer effectively illustrates how time, origin and dispersal history together explain the patterns we observe. *Ficedula buruensis* is an old species that has been on Buru/Seram for several million years whereas *Ficedula hyperythra* is a recent coloniser. Importantly, however, *Ficedula* represents a Palearctic group of birds that have had markedly different colonisation histories compared to taxa with Australasian origins. In many cases, Palearctic/montane Asian species have colonised the Indo-Pacific islands from mountaintop to mountaintop as opposed to Australasian taxa that tend to colonise through the lowlands. This information comes from broader phylogenetic analyses that we briefly discuss in the text.

The second is that owing to this variance across the histories of the different taxa, really large sample sizes are required to make generalizable conclusions. This is very challenging to accomplish, but it has been done (for example, in studies of dispersal ability and differentiation levels in South American birds). Here, the comparison of just one mountain gap with just one island gap (each with its own set of non-comparable histories), and the relatively modest number of taxa compared for each, limits the power of this study to identify generalizable patterns.

RESPONSE: Yes, more individuals would be preferred, but as stated by the reviewer above, and also by reviewer 3, the field efforts for this project have been enormous. We do not consider this study the final word, but as a first step in the age of genomics to understand how biotas have built up in this part of the world and beyond.

In first reading the abstract of this paper, I was most drawn to its potential to address the possibility of in situ montane speciation via selection across ecological gradients. Some of the authors of the present study have impressive histories of work on this somewhat controversial topic.

Unfortunately, I think they are making a basic mistake in how they analyze their elevational data along these gradients. There is a vast recent literature on these processes showing that divergence-

with-gene-flow does not manifest as a genome-wide phenomenon (at least, not until the taxa are so differentiated that gene flow has ceased). Instead, it proceeds via selection acting to create small 'islands of differentiation' within the genome against a background of negligible differentiation. Therefore, even if ecological gradients were driving in situ divergence in these montane birds, one would not expect to even glimpse it in the kinds of gross whole-genome *Fst* or STRUCTURE comparisons that this paper presents. For other reasons, I agree with the authors that this kind of in situ montane speciation is likely to be rare (at best), but I do not think that their gross differentiation approach is a viable method for testing this hypothesis. As an aside, the sample sizes for these within-taxon elevational comparisons are also not sufficient.

RESPONSE: We agree that when we split the samples for a population on a single gradient into elevational groups, we are getting down to very small sample sizes and that the parapatric speciation analyses are not very powerful. For *Origma robusta* we did initially investigate if individuals at 1,700 m (N=2) and 3,700 m (N=2) had differentially fixed alleles. In the end, however, we decided against pursuing this approach, as we accept that a 2 by 2 comparison lacks sufficient statistical power. In the revised version of the manuscript, we have substantially toned down the conclusions with respect to parapatric speciation, explicitly mentioning (in the results and in the discussion) that analyses with more individuals testing for islands of differentiation will be necessary to properly address this question.

I caution the authors for potentially reading too much into the fastsimcoal2 plots of demographic parameters. fastsimcoal has been a workhorse for this kind of inference for almost a decade, partly because there are few good analytical alternatives, but these models have real limitations. What are the error estimates on these parameters? An entry point into this topic is the 2017 book chapter by Salmona et al. Inferring Demographic History Using Genomic Data.

RESPONSE: We certainly agree that demographic analyses such as fastsimcoal2 should be interpreted with some caution. However, fastsimcoal2 does still represent one of the most powerful approaches to determine demographic parameters. We highlight that error margins for the preferred models are presented in Supplementary table 3. Furthermore, Rasmus Heller (one of the co-authors of the book chapter mentioned) is a colleague of ours at the University of Copenhagen and at the outset of this study we were advised that this approach still represents the best way of analysing our data to address the questions of interest.

Reviewer #2 (Remarks to the Author):

I enjoyed reading the paper and found it to be another useful contribution of molecular population genomics and demographics to understanding the dynamics of montane bird evolution in the Indo-Pacific. Along with paper like the recent Garg et al paper in BMC E and E, this is developing into a model suite of papers and I thank and congratulate the authors.

RESPONSE: Many thanks for the nice words.

Most of my comments are on the attached pdf. Some key issues are:

I found the aims a little confused in so far as selection and adaptation seemed to come and go in various parts of the paper. Perhaps that could be clarified.

RESPONSE: We are not explicitly addressing selection and have revised the text to omit any use of the word. Hopefully these efforts have removed any further confusion.

The paper is a little lop-sided in its discussion of the New Guinean results relative to the Moluccan results. That is understandable as the story seems more complicated in New Guinea. Yet after I had

finished reading the paper, I went back into the earlier sections looking for some more details about the Moluccas and New Guinea together. In particular, the ages of the mountains and the sea/land gaps in the Moluccas and New Guinea need to be clearer and easier to compare. Are the Moluccan results just being ascribed (except in *Ficedula* I think) to vicariance either side of the ocean gap? Is that why New Guinea needs more detailed discussion? Maybe lead the reader a little more kindly through that.

RESPONSE: Yes, the discussion is slightly lop-sided, but as you mention that is because we have included more population-pairs from New Guinea and the story is more complex. We have made the age of the mountains (and the barrier) in New Guinea explicit in the introduction. As for the last point, we ascribe diversification not so much to vicariance but more to the fact that the two populations have quickly embarked on different evolutionary trajectories following their colonisation of the islands.

Patton and Smith 1992 as described in the comments on the pdf is worth a look I suspect.

Response: Indeed a nice paper that we now cite it in the introduction.

Lots of long sentences. Break them up more, please! I have suggested some ways to tackle this.

RESPONSE: We have done our utmost to weed out long and convoluted sentences. Thank you for pointing out specific sentences in the pdf comments, these, in addition to some others have been revised.

Reviewer #3 (Remarks to the Author):

This paper takes advantage of a favorable geographic situation to address many basic debated questions of evolution and biogeography: questions of speciation, genetic differentiation, dispersal, taxon cycles, population sizes, and others.

The geographic situation is the mountains of New Guinea and of the Moluccan islands, where the authors measured and compared genetic differentiation across two different barriers: the lowland barrier between New Guinea's Central Range and its Huon Mountains, and the ocean barrier between Moluccan islands.

At the outset of this review, I'll mention a strength of this study that the authors modestly never mention and that is likely to make it unique: the extreme physical, logistic, and permitting obstacles to doing research in the mountains of Buru, Seram, New Guinea's Central Range, and New Guinea's Huon Mts. To accomplish this study in just one of those four mountains would have been difficult; to accomplish it in all four mountains, difficult to the extreme.

RESPONSE: We could not agree more.

The only significant problem that I find in this important paper is to present clearly the diverse sets of results. Hence my review will consist of my summary of the results, followed by my suggestions for presentation.

SUMMARY OF RESULTS. The authors address nine questions, and reach nine sets of conclusions:

1. Genetic differentiation of mountain birds is generally greater across the ocean barrier than across the lowland barrier.

2. Within this generalization, there are some differences among species within the same fauna.
3. Differentiation is greater between populations of montane species than between populations of lowland species.
4. Differentiation is greater for montane species with higher altitudinal floors, in New Guinea but not in the Moluccas.
5. There were marked fluctuations in population sizes during the Pleistocene.
6. There is negligible differentiation along elevational gradients, hence no evidence for sympatric speciation in this study.
7. The montane populations studied originated by jumping between mountains. There is only limited support for origins by push-pull, as in taxon cycles.
8. The direction of colonization is predominantly from a larger mountain range to a smaller mountain range.
9. Dispersal can be estimated from genetic evidence. I consider this finding a big deal about which the authors are being too modest. Knowledge of dispersal is fundamental to understanding biogeography. We know a lot about bird dispersal in Europe and North America, from the observations of millions of bird-banders. Such information is almost non-existent for New Guinea. This paper's dispersal estimates from genetic evidence are currently unique for the New Guinea region.

SUGGESTIONS FOR FRAMING THE PRESENTATION OF THESE NINE QUESTIONS AND CONCLUSIONS MORE CLEARLY.

In the first paragraph of the introduction, succinctly mention several of these major detailed questions, as providing the motivation for this paper.

RESPONSE: This is a good idea. In the revised version we explicitly mention the questions at the end of the opening paragraph.

In the second paragraph of the introduction, explain your favorable test situation for answering these questions: comparison of the ocean barrier between Buru and Seram with the lowland barrier between New Guinea's Central Range and Huon Mts., as those barriers affect genetic differentiation of 18 montane forest bird species.

RESPONSE: We now clarify how the answering the key questions can inform about the underlying mechanisms.

In the last paragraph of the introduction, succinctly list all nine questions.

RESPONSE: The key questions are now mentioned in the first paragraph as suggested above.

In the first paragraph of the discussion, orient readers by succinctly listing (one sentence each) your conclusions about these nine questions, before plunging into details of your conclusions. For example, you could modify my draft "summary of results" above.

RESPONSE: Thank you very much. We have done just that.

OTHER SUGGESTIONS FOR IMPROVING CLARITY

Some important concepts or acronyms aren't defined at the first mention in the text. Please define them: parapatric speciation in situ (line 130), FST (line 183), STRUCTURE (line 187), $K = 1,2,3,\dots$ (line 213).

RESPONSE: We have added further definitions of parapatric speciation, FST and STRUCTURE immediately following their first use in the main text.

In addition to those definitions in the text at first mention, gather and repeat all definitions of acronyms in a footnote near the beginning of the paper.

RESPONSE: We do believe that all terms are well defined on first mention. As far as we are aware footnotes are not used by Nature Communications, but if the editor finds it necessary to include these, we are happy to follow her advice.

SPECIFIC SUGGESTIONS

Line 243: the expression "minimum elevation" is unclear. Say instead "altitudinal floor," and add a phrase to explain.

RESPONSE: we have amended the text accordingly.

Line 135: the two New Guinea mountain ranges are not separated by a river barrier as stated in this line, but instead by a lowland barrier. The Ramu and Markham Rivers themselves constitute only a small fraction of this lowland barrier's width.

RESPONSE: Agreed, Thanks for catching this one.

Line 216: say that Mt. Scratchley is in the Central Range as is Mt. Wilhelm, but 400 kilometers to the east, and separated by a much weaker lowland barrier than that between Mt. Wilhelm and the Huon Mts. Otherwise, readers won't understand the significance of Mt. Scratchley: it's not enough for readers to guess the significance from the figured map.

RESPONSE: That is a valid point and we have amended the text accordingly.

Lines 437 - 470, section on jumping. This section argues that montane species colonize other mountains by jumping, not by push-pull. That may be true, but mention that the ORIGIN of a montane species is a separate question; that question of the origin of montane species is the question addressed by explanations involving taxon cycles and push-pull.

RESPONSE: This is a very valid point and we have revised this section markedly to make it clear that the origin of montane species is a separate question.

Reviewer: Jared Diamond, UCLA

Peer Review comments, second round review –

Reviewer #2 (Remarks to the Author):

I thank the authors for attending to my comments and those of other reviewers.

Reviewer #4 (Remarks to the Author):

Interest of the study

In this work, the authors have compiled an impressive dataset to compare differentiation across two main types of barriers in the New Guinea region for several bird species. They highlight the role of physical barriers, elevation and carrying capacities on the historical population dynamics. I enjoyed reading the article. I also read comments from previous reviewers and I am mostly happy with the answers. This study is the result of hard fieldwork and represents a massive sequencing effort. The fact that “only” three comparisons could be included is balanced, in my opinion, by the importance of the resource for future studies on this iconic biodiversity hotspot.

Major general comments

I think that presenting more clearly the results from demographic models could help partially address reviewer 1’s comment on “historical idiosyncrasies”. The authors discuss quite a lot their FST estimates and PSMC curves. However, in my opinion, the estimates they obtain from their fastsimcoal2 models are possibly more informative (keeping in mind the limitations of such models). Setting local adaptation/ecological speciation aside, Fst will mostly increase (on average) with increasing divergence times and decreasing gene flow. You could for example test for the effects of barriers on dispersal by comparing estimates of the effective number of migrants N^*m . You could test for the effects of historical changes in environmental conditions by examining fluctuations in N (this part is already quite clear). At last, you can also examine the historical effects of barriers, for example by comparing divergence times between populations with orogenesis and the emergence of mountain ranges.

Technical comments

PSMC is starting to be a bit outdated, and is designed to work with a single diploid genome. However, you have more than a single diploid individual per population and can make a better use of the information at hand. I would suggest using an approach such as SMC++ (<https://github.com/popgenmethods/smcpp>), which does not require phasing. Using such an approach might give you more resolution over recent times (which I assume would be useful to examine the dynamics since the LGM).

Another possibility could be using MSMC-IM (<https://github.com/wangke16/MSMC-IM>) and compare its results with the ones from fastsimcoal2. The issue is that MSMC-IM requires phased data. I am somewhat reluctant to apply statistical phasing over relatively small sample sizes, so I would pay close attention to deviations between SMC++ and MSMC-IM estimates, particularly over recent times. If these results agree, it suggests phasing may not cause too much trouble. You would have an interesting picture of how connectivity and population sizes have changed for all these species over time. I leave that option to the authors, as it is more exploratory, but I would be very curious of the results.

About using migration rates: I think that it would be good to present the values for N^*m (“effective number of migrants per generation”) rather than m only. Ultimately, if you are interested in dispersal, you also want to estimate the number of migrants crossing the barriers. This estimate also accounts for differences in effective population sizes: for example, a low m and a low N imply much more divergence than a low m with a high N . It would also please some population geneticists.

A minor point: why did you use STRUCTURE and not a faster option, such as ADMIXTURE? There is nothing wrong in using it, and the results should hold, but it may strike the reader as odd.

I hope you find these comments constructive. The amount of work is really impressive.

Best regards

Reviewer #4 (Remarks to the Author):

Interest of the study

In this work, the authors have compiled an impressive dataset to compare differentiation across two main types of barriers in the New Guinea region for several bird species. They highlight the role of physical barriers, elevation and carrying capacities on the historical population dynamics. I enjoyed reading the article. I also read comments from previous reviewers and I am mostly happy with the answers. This study is the result of hard fieldwork and represents a massive sequencing effort. The fact that “only” three comparisons could be included is balanced, in my opinion, by the importance of the resource for future studies on this iconic biodiversity hotspot.

RESPONSE: Thank you very much for the kind words. It has taken years of fieldwork to collect the data and more years of lab and analytical work to get to this stage. So, we are quite pleased that this is recognised. THANK YOU!

Major general comments

I think that presenting more clearly the results from demographic models could help partially address reviewer 1’s comment on “historical idiosyncrasies”. The authors discuss quite a lot their FST estimates and PSMC curves. However, in my opinion, the estimates they obtain from their fastsimcoal2 models are possibly more informative (keeping in mind the limitations of such models). Setting local adaptation/ecological speciation aside, Fst will mostly increase (on average) with increasing divergence times and decreasing gene flow.

RESPONSE: We agree and have tried to keep a balanced discussion that includes both Fst and the fastsimcoal2 results. However, in the revised version we now specifically report the number of effective migration (dispersal) events in the discussion section “*Does barrier strength predictably influence differentiation of disjunct populations?* “. Moreover, as suggested by the reviewer below, we report the number of dispersing individuals per generation rather than only refer to the migration rates.

You could for example test for the effects of barriers on dispersal by comparing estimates of the effective number of migrants N^*m .

RESPONSE: As mentioned above, we agree that it is a good idea to also calculate the effective number of migrants. We have done this and added the values to table 1 and also mention these dispersing individuals explicitly in the discussion.

You could test for the effects of historical changes in environmental conditions by examining fluctuations in N (this part is already quite clear).

RESPONSE: We agree that this is well-covered in the manuscript.

At last, you can also examine the historical effects of barriers, for example by comparing divergence times between populations with orogenesis and the emergence of mountain ranges.

RESPONSE: This is something we have put a lot of thought into, we finally landed on a compromise in which we largely focus on whether populations are diverged or not. Fastsimcoal (Table 1) estimates relatively old divergence times even for panmictic populations (notably *Melilestes* T DIV=1.4 My and *Toxorhamphus* T DIV=1.9 My). Thus, we are cautious with our interpretations of these divergence time estimates. Moreover, the time of emergence of specific mountain regions are surrounded by substantial uncertainty. We know that the Central Range is roughly 5-10 My and that the Huon mountains are a great deal younger (likely 2-3 My). This is

mentioned in the text, but other than that we cannot make any strong predictions based on geological events and have therefore refrained from it.

Technical comments

PSMC is starting to be a bit outdated, and is designed to work with a single diploid genome. However, you have more than a single diploid individual per population and can make a better use of the information at hand. I would suggest using an approach such as SMC++ (<https://github.com/popgenmethods/smcpp>), which does not require phasing. Using such an approach might give you more resolution over recent times (which I assume would be useful to examine the dynamics since the LGM).

RESPONSE: We are aware of these other methods you mention. In the initial stages of the project, we did some tests and the impression was that it would not provide much additional insight, which is not unexpected since alternative models of molecular evolution or statistical frameworks (i.e., MSMC or SMC++) are not radically different to PSMC. In the end, we opted for PSMC for various reasons as we believe that it is the best approach given the data we have. For each species, these data consist of (i) one individual for which we generated a de novo assembly (ii) an additional 5-10 individuals from different populations that were re-sequenced. Our approach is the same used in the study of Nadachowska-Brzyska for *Ficedula* flycatcher populations, which showed that re-sequenced data can also be used for PSMC analyses. Indeed, on inspection of our PSMC plots, the Ne curves obtained using re-sequenced data fit the pattern observed in the Ne curves obtained with whole genome sequencing data. The additional advantage of using all the resequenced individuals is that we can see not only how the effective population sizes of a species changes through time, but also how effective population sizes of specific populations change through time. We believe that PSMC is the best approach for our data since we only generated one de novo assembly per species and do not think resequenced data have been used with SMC++. On the other hand, PSMC clearly works with both data types and can actually be combined in the same figure and discussed together.

That being said, exploring different approaches to estimate demographic parameters is very exciting and something we intend to explore further with a broader sampling of taxa across the Indo-Pacific.

Another possibility could be using MSMC-IM (<https://github.com/wangke16/MSMC-IM>) and compare its results with the ones from fastsimcoal2. The issue is that MSMC-IM requires phased data. I am somewhat reluctant to apply statistical phasing over relatively small sample sizes, so I would pay close attention to deviations between SMC++ and MSMC-IM estimates, particularly over recent times. If these results agree, it suggests phasing may not cause too much trouble. You would have an interesting picture of how connectivity and population sizes have changed for all these species over time. I leave that option to the authors, as it is more exploratory, but I would be very curious of the results.

RESPONSE: This is a good point and we did try phasing as this opens up for the possibility of using a number of other approaches, not only MSMC but any software based on haplotypes like iHS or XP-EHH. The problem that we have is that all our genomes are not assembled at the chromosome level but only at scaffold level. So, our genomes consist of hundreds of thousands of tiny contigs and scaffolds. We did try phasing using the SHAPEIT software but resolution was not good enough. Phasing without a better assembly seems risky at best as you do not know the order of the contigs.

Clearly, thorough testing of these different approaches to estimate population sizes through time is exciting but it is very time-consuming and, we believe, beyond the scope of this study.

About using migration rates: I think that it would be good to present the values for $N*m$ (“effective number of migrants per generation”) rather than m only. Ultimately, if you are interested in dispersal, you also want to estimate the number of migrants crossing the barriers. This estimate also accounts for differences in effective population sizes: for example, a low m and a low N imply much more divergence than a low m with a high N . It would also please some population geneticists.

RESPONSE: That is a very good point. We have added the number of effective migrants to Table 1 and also explicitly mention some of the relevant estimates in the discussion (see above).

A minor point: why did you use STRUCTURE and not a faster option, such as ADMIXTURE? There is nothing wrong in using it, and the results should hold, but it may strike the reader as odd.

RESPONSE: For all species, we investigated population substructuring with both STRUCTURE and ADMIXTURE. The results were virtually identical except that STRUCTURE detected some subtle substructuring that ADMIXTURE did not. In the end, we decided to only show the results of STRUCTURE.

I hope you find these comments constructive. The amount of work is really impressive.

RESPONSE: Again, thank you very much.

Best regards,
Yann Bourgeois

Peer Review comments, third round review –

Reviewer #4 (Remarks to the Author):

The authors have done a very good job addressing my comments and I support the publication of the current version. I believe this study will generate a lot of interest, and will appeal to the broad readership of Nature Communications. The study represents a massive sampling and sequencing effort, and paves the way for further studies on this historically important biodiversity hotspot.